# Multi-dataset Joint Pre-training of Emotional EEG Enables Generalizable Affective Computing

**Qingzhu Zhang**[1]    **Jiani Zhong**[2]    **Zongsheng Li**[3]    **Xinke Shen**[1*]    **Quanying Liu**[1*]

[1]Department of Biomedical Engineering, Southern University of Science and Technology, Shenzhen, China
[2]School of Data Science, University of California, San Diego, La Jolla, CA, USA
[3]School of Science and Engineering, The Chinese University of Hong Kong, Shenzhen, China

`12432852@mail.sustech.edu.cn`, `{121090846, zongshengli}@link.cuhk.edu.cn`
`{shenxk, liuqy}@sustech.edu.cn`

## Abstract

Task-specific pre-training is essential when task representations diverge from generic pre-training features. Existing task-general pre-training EEG models struggle with complex tasks like emotion recognition due to mismatches between task-specific features and broad pre-training approaches. This work aims to develop a task-specific multi-dataset joint pre-training framework for cross-dataset emotion recognition, tackling problems of large inter-dataset distribution shifts, inconsistent emotion category definitions, and substantial inter-subject variability. We introduce a cross-dataset covariance alignment loss to align second-order statistical properties across datasets, enabling robust generalization without the need for extensive labels or per-subject calibration. To capture the long-term dependency and complex dynamics of EEG, we propose a hybrid encoder combining a Mamba-like linear attention channel encoder and a spatiotemporal dynamics model. Our method outperforms state-of-the-art large-scale EEG models by an average of 4.57% in AUROC for few-shot emotion recognition and 11.92% in accuracy for zero-shot generalization to a new dataset. Performance scales with the increase of datasets used in pre-training. Multi-dataset joint pre-training achieves a performance gain of 8.55% over single-dataset training. This work provides a scalable framework for task-specific pre-training and highlights its benefit in generalizable affective computing. Our code is available at `https://github.com/ncclab-sustech/mdJPT_nips2025`.

## 1    Introduction

In domains like neuroscience, data is often limited, heterogeneous, and intrinsically linked to specific experimental tasks. This makes task-specific pre-training a particularly promising paradigm, especially for modalities such as EEG. Unlike generic large-scale pre-training, which aims to learn broad, task-agnostic representations, task-specific pre-training focuses on inducing representations tailored to a coherent problem class, incorporating task-relevant inductive biases and domain structure. Recent EEG foundation models (1; 2; 3; 4; 5; 6) adopt generic pre-training by aggregating heterogeneous datasets across multiple tasks. While effective on broad benchmarks like sleep staging or anomaly detection, such models often struggle with tasks that involve more complex and nuanced neural representations (1). This performance gap stems from severe inter-task representation mismatch and the dilution of task-relevant signals during large-scale heterogeneous pre-training.

---

[*]Corresponding authors

39th Conference on Neural Information Processing Systems (NeurIPS 2025).

In contrast, task-specific multi-dataset pre-training, which leverages multiple datasets targeting the same cognitive function, offers a focused alternative. EEG-based emotion recognition is a natural candidate for this paradigm: it has growing availability of labeled datasets (e.g., SEED (7), DEAP (8), FACED (9)) and strong practical value in affective computing and BCI applications. However, developing such a framework faces several technical challenges: large inter-dataset distribution shifts, heterogeneous EEG montages, inconsistent emotion category definitions, and substantial inter-subject variability. Existing approaches typically rely on one-to-one dataset adaptation and subject-specific fine-tuning (10; 11; 12; 13; 14; 15) (Table 1), limiting their generalization to unseen datasets or emotion categories (Fig. 1). These limitations call for a more principled task-specific pre-training strategy that aligns across datasets without requiring extensive labels or per-subject calibration.

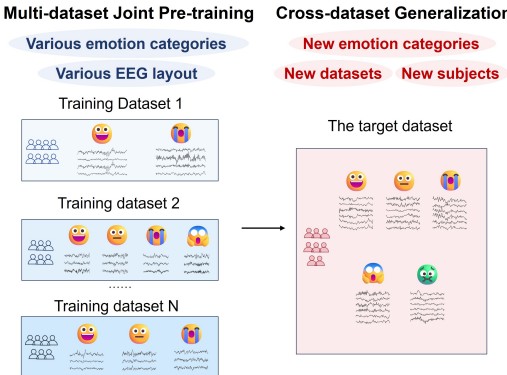

Figure 1: Cross-dataset generalization through multi-dataset pre-training. (left) Challenges of task-specific EEG pre-training. (right) Challenges of inference.

Inspired by the evidence that EEG signals exhibit similar second-order statistical properties (such as correlations or covariance) across datasets, which can be aligned through simple transformations (16; 17), we develop a label-free, task-specific pretraining framework that learns transferable emotion representations by aligning statistical patterns, while maintaining robustness across both datasets and subjects. Our framework harmonizes cross-dataset feature structures and captures discriminative neural dynamics through tailored spatiotemporal modeling (Fig. 2). Our main contributions are as follows:

- **Multi-dataset joint pre-Training (mdJPT)**: We develop a scalable multi-dataset pre-training framework for EEG-based emotion recognition, called mdJPT. We validated the superior performance of mdJPT over generic EEG foundation models in cross-dataset generalization.

- **Cross-dataset alignment (CDA) loss**: We introduce a novel CDA loss that aligns second-order feature statistics across datasets, effectively mitigating inter-dataset and inter-subject distribution shifts. This enables zero-shot generalization to new emotion categories and unseen datasets.

- **Hybrid spatiotemporal encoder**: We design a Mamba-like linear attention (MLLA)-based encoder (18) augmented with spatial transition convolutions and dynamic attention, enabling robust extraction of emotion-related long-term EEG dynamics and inter-channel dependencies.

## 2 Method

### 2.1 Task-specific multi-dataset joint pre-training

Our work aims to develop a unified framework for task-specific multi-dataset EEG training and cross-dataset generalization (Fig. 2). We introduce two complementary alignment losses: the Covariance Distribution Alignment (CDA) loss and the Inter-Subject Alignment (ISA) loss. The CDA loss aligns global second-order statistics (i.e., inter-channel covariance structures) across subjects and datasets (Fig. 2B). The ISA loss complements the CDA loss by a more fine-grained inter-subject alignment (Fig. 2C). For the EEG encoder, we aim to design a physiologically plausible and computationally efficient architecture. It consists of an MLLA-based channel encoder (Fig. 2D) and a spatiotemporal dynamics model (Fig. 2E). The former captures long-term temporal dynamics from each EEG channel, and the latter integrates information across channels and time, enabling the model to learn coordinated patterns of brain activity.

Our approach is validated under two settings: (i) cross-dataset subject-independent (few-shot) classification and (ii) zero-shot generalization. The few-shot classification consists of three stages:

multi-dataset joint pre-training, classifier fine-tuning, and testing. In the first stage, an EEG encoder is pre-trained on multiple datasets using CDA and ISA loss. In the second stage, the encoder is frozen, and a classifier is fine-tuned on the extracted representations from a few labeled subjects in the target dataset. In the final stage, the trained model is evaluated on the remaining subjects of the target dataset (Fig. 2A). The zero-shot generalization setting has no fine-tuning stage, with the pre-trained model directly generalized to a new dataset.

Table 1: Comparison with existing cross-dataset emotion recognition methods.

| Methods | Multi-dataset joint pre-training? | Generalizable to new emotion categories? | Generalizable to new subjects w/o finetuning? |
|---|:---:|:---:|:---:|
| PESD (13) | × | × | × |
| E$^2$STN (14) | × | × | × |
| JCFA (15) | × | × | × |
| Imtiaz & Khan (10) | × | × | × |
| SCMM (12) | × | ✓ | × |
| DBDG (11) | × | × | ✓ |
| **mdJPT (Ours)** | ✓ | ✓ | ✓ |

## 2.2 The EEG encoder

**MLLA channel encoder**. The MLLA architecture is employed for modeling long-range dependencies in EEG. Preprocessed EEG data with heterogeneous electrode layouts from different datasets are first interpolated to a standardized 60-channel configuration based on the 10-20 International System (19) to ensure spatial consistency. The interpolated EEG input can be denoted as $x \in \mathbb{R}^{C \times L}$. Then we split EEG signals into overlapped segments. Specifically, for EEG time series of the $c^{th}$ channel, $x^{(c)} \in \mathbb{R}^{1 \times L}$, we divide them into overlapping strided patches (Fig. 2D), with patch length of $P$ and the stride length of $S$. The sequence of patches are denoted as $x_p^{(c)} \in \mathbb{R}^{N_1 \times P}$, where $N_1 = (\lfloor \frac{L-P}{S} + 1 \rfloor)$ indicates the number of patches. Patches derived from each channel are fed independently into a MLLA (18) encoder. For each $x_p^{(c)}$, the MLLA backbone then provide results $\hat{x}_p^{(c)} \in \mathbb{R}^{N_1 \times K_1}$, where $K_1$ is the output dimension of MLLA encoder. Each EEG segment $x$ produces an output $\hat{x} \in \mathbb{R}^{C \times N_1 \times K_1}$. The MLLA encoder includes an input gate, a linear attention module, and a forget gate. The input gate contains a linear layer and a convolution layer for temporally local processing. With strided patches as inputs, the linear layer is functionally similar to strided convolution. The linear attention operator can model global dependencies, with its output multiplied by the output of a linear-layer-comprised forget gate to filter salient patterns.

**The spatialtemporal dynamics model**. EEG signals exhibit complex spatiotemporal dynamics, with temporally varying spatial co-activations across electrodes. To capture these inter-channel dependencies, we introduce a spatiotemporal dynamics model that employs local attention to highlight salient spatial-transition patterns at each time step (Fig. 2E). Firstly, we pass $\hat{x}$ through a trainable linear spatial projector $W_s \in \mathbb{R}^{C \times C}$, mapped to a latent space for covariance alignment (see section 2.3): $p = W_s \hat{x}$, where $p \in \mathbb{R}^{C \times N_1 \times K_1}$ and $p^{(d)} \in \mathbb{R}^{C \times N_1}, d = 1, 2, ..., K_1$ represents a slice in $p$. A spatial transition convolution is then employed, utilizing convolutional kernels of size $C \times L_1$ to capture spatial variation patterns across multiple time steps $L_1$. The convolutional kernels use dilations with various temporal intervals, enabling the extraction of EEG variation patterns across different time scales, yielding latent spatiotemporal representations $h^{(1)} \in \mathbb{R}^{K \times N_1}$ ($K$ is the number of hidden dimensions and $N_1$ is the number of time steps. Subsequently, local attention is adopted to estimate time-varying importance weights on spatiotemporal representations following (20). A depthwise one-dimensional temporal convolution, an average pooling over the temporal dimension, and a pointwise convolution mixing different channels are adopted to estimate the attention weights based on latent spatiotemporal representations. The attention weights are multiplied with the spatiotemporal representations to yield the weighted latent patterns $h^{(2)} \in \mathbb{R}^{K \times N_1}$. See Appendix B for details of the spatiotemporal dynamics model.

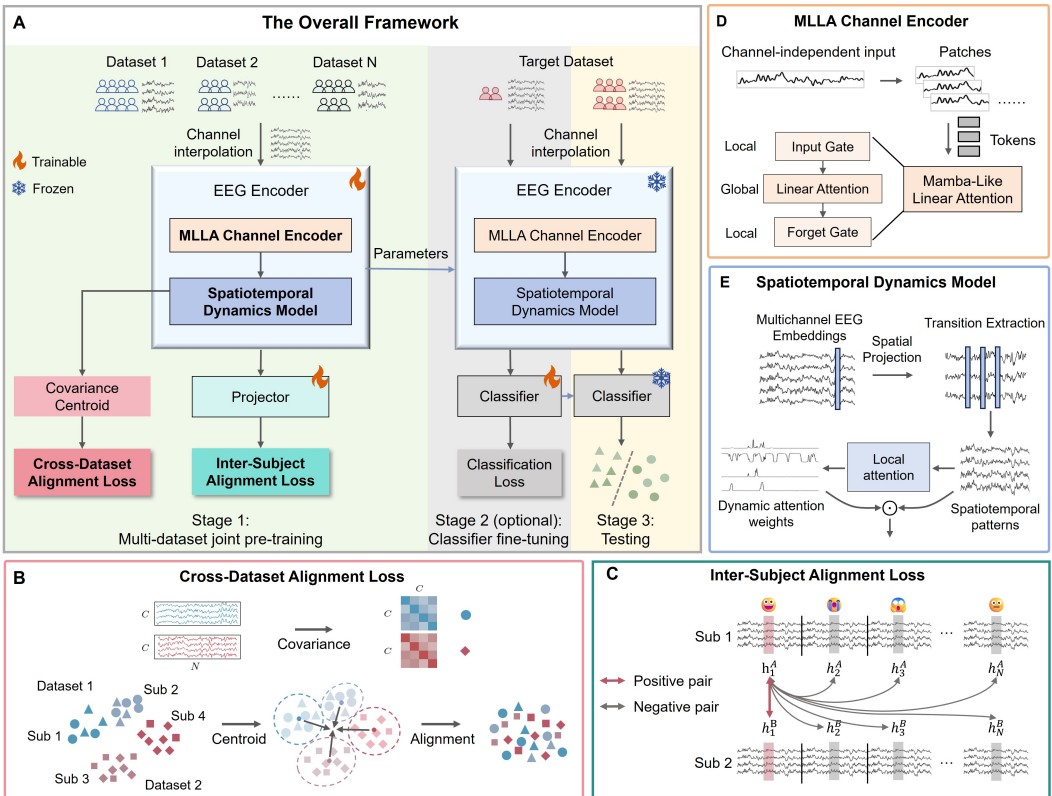

Figure 2: The overall framework, model architectures, and loss designs of mdJPT. **A**) The overall framework. Our approach comprises three stages: multi-dataset pre-training, optional classifier fine-tuning, and testing. The EEG encoder integrates an MLLA channel encoder with a spatiotemporal dynamics model. Pre-training combines CDA and inter-subject alignment (ISA) losses. The resulting model supports either zero-shot generalization or fine-tuning with a lightweight classifier on limited target data. Final evaluation is performed on held-out subjects. **B)** The cross-dataset alignment loss. The covariance is calculated from the latent representations, and covariance centroids of subjects from different datasets are aligned. **C**) The inter-subject alignment loss. EEG samples from two subjects corresponding to the same time segment within a trial are regarded as positive samples, and those corresponding to different trials are regarded as negative samples. A contrastive loss is employed to pull together the positive samples and push apart the negative samples. **D**) MLLA channel encoder. A channel-independent Mamba-like linear attention encoder is employed to process the time-slided patches of each EEG channel. **E**) Spatiotemporal dynamics model. The multichannel EEG embeddings are processed by spatial projections and transition extraction to obtain spatiotemporal EEG patterns, then a local attention is employed to estimate dynamic attention weights on these patterns.

## 2.3 Cross-dataset alignment loss

To mitigate the discrepancy between datasets and subjects, we design the CDA Loss based on covariance centroid alignment. Specifically, we compute the subject centroid of the covariance matrices derived from the EEG latent space ($p$) in a training batch. The loss minimizes the Euclidean distance across centroids of different datasets or subjects, thereby reducing distributional discrepancies in EEG feature representations across datasets.

Each training batch contains EEG data from $2M$ subjects, where $M$ is the number of datasets involved in pre-training. $v_m$ samples are extracted from each subject, where $v_m$ is the number of trials in dataset $m$. The CDA loss needs to align the covariance centroid of the $2M$ subjects. The covariance of the spatial projection layer's output $p$ is calculated:

$$\Sigma_{s,v}^{(d)} = \frac{1}{N_1 - 1} \left( p_{s,v}^{(d)} - \bar{p}_{s,v}^{(d)} \right) \left( p_{s,v}^{(d)} - \bar{p}_{s,v}^{(d)} \right)^\top, d = 1, 2, ..., K_1 \tag{1}$$

where $p_{s,v}^{(d)} \in \mathbb{R}^{C \times N_1}$ is the hidden representation from trial $v$, subject $s$. $\bar{p}_{s,v}^{(d)} \in \mathbb{R}^C$ is the temporal average of $p_{s,v}^{(d)}$. The covariance of all samples from each subject is averaged to obtain the subject-level covariance centroids:

$$\mathbf{\Gamma}_s^{(d)} = \frac{1}{v_m} \sum_{v=1}^{v_m} \mathbf{\Sigma}_{s,v}^{(d)}, d = 1, 2, ..., K_1 \tag{2}$$

yielding $2M$ subject-wise centroids $\{\mathbf{\Gamma}_1^{(d)}, ..., \mathbf{\Gamma}_{2M}^{(d)}\}$ per dimension $d$. The alignment loss measures pairwise Euclidean distances between all $2M$ centroids:

$$\mathcal{L}_d = \sum_{m=1}^{2M} \sum_{n=m+1}^{2M} \left\| \mathbf{\Gamma}_m^{(d)} - \mathbf{\Gamma}_n^{(d)} \right\|_F^2, d = 1, 2, ..., K_1 \tag{3}$$

where $\|\cdot\|_F$ denotes the Frobenius norm. The loss of all $K_1$ dimensions is summed, and the logarithm is taken to yield the final Cross-Dataset Alignment Loss:

$$\mathcal{L}_{CDA} = log(\sum_{d=1}^{K_1} \mathcal{L}_d + 1). \tag{4}$$

### 2.4 Inter-subject alignment loss

A contrastive-learning-based ISA loss is employed to mitigate inter-subject discrepancy. In the emotional EEG datasets, subjects are required to watch a series of emotional video stimuli. ISA loss aims to distinguish whether EEG segments from two subjects correspond to the same stimulus or different stimuli. This approach pulls closer the representations from different subjects' similar states and pushes apart the representations of different emotion states (21). ISA loss enables the model to learn potential emotion-relevant EEG representations without requiring explicit emotion labels, thereby overcoming the challenge of inconsistent emotion label categories across different pre-training datasets.

In a training batch, a positive pair is formed by pairing EEG segments from two different subjects who were exposed to the same emotional stimulus. Samples corresponding to mismatching emotional stimuli form the negative pairs. The contrastive loss is calculated on the two subjects from each dataset and summed over all datasets. We denote the samples in a dataset $m$ as: $\{x_{m,v}^s | v = 1, 2...v_m, s \in A, B\}$. For a sample $x_{m,i}^A$, $x_{m,i}^B$ forms a positive pair with it, and the other $2(N-1)$ samples $\{x_{m,j}^s | j = 1, 2...v_m, j \neq i, s \in A, B\}$ form negative pairs.

The contrastive loss is calculated based on the output of a projector. The projector contains two convolutional layers over the temporal dimension. It receives the output of the EEG encoder $h^{(2)}$ and projects it to $h \in \mathbb{R}^{K' \times N_1}$. The similarity between two samples is defined as:

$$sim(h_{m,i}^A, h_{m,i}^B) = \frac{h_{m,i}^A \cdot h_{m,i}^B}{\|h_{m,i}^A\| \|h_{m,i}^B\|} \tag{5}$$

The normalized temperature-scaled cross-entropy loss (22) is calculated based on the sample similarity:

$$l_{m,i}^A = -\log \left[ \frac{\exp(sim(h_{m,i}^A, h_{m,i}^B)/\tau)}{\sum_{j=1}^N \mathbb{1}_{[j \neq i]} \exp(sim(h_{m,i}^A, h_{m,j}^A)/\tau) + \sum_{j=1}^N \exp(sim(h_{m,i}^A, h_{m,j}^B)/\tau)} \right] \tag{6}$$

where $\mathbb{1}_{[j \neq i]} \in \{0, 1\}$ is an indicator function. It is set to 1 if $j \neq i$. By minimizing the loss function, the model will increase the similarity between $h_{m,i}^A$ and $h_{m,i}^B$ in contrast to all other possible sample pairs involving $h_i^A$. The ISA loss for a training batch is:

Table 2: Summary of emotion EEG datasets used for cross-dataset evaluation.

| Dataset | Sampling Rate (Hz) | #Subjects | #Trials | #Emo. Classes | EEG Device (#Channels) |
|---------|--------------------|-----------|---------|---------------|------------------------|
| SEED | 1000 | 15 | 45 | 3 | ESI NeuroScan System (62) |
| SEED-IV | 1000 | 15 | 72 | 4 | ESI NeuroScan System (62) |
| SEED-V | 1000 | 16 | 45 | 5 | ESI NeuroScan System (62) |
| SEED-VII | 1000 | 20 | 80 | 7 | ESI NeuroScan System (62) |
| FACED | 250/1000 | 123 | 28 | 2 / 9 | NeuSen.W32, Neuracle (32) |
| DEAP | 512 | 32 | 40 | 2 | Biosemi ActiveTwo system (32) |

$$\mathcal{L}_{ISA} = \sum_{m=1}^{M} \left( \sum_{i=1}^{v_m} l_i^A + \sum_{i=1}^{v_m} l_i^B \right) \tag{7}$$

The total loss in pre-training is defined as:

$$\mathcal{L} = \mathcal{L}_{ISA} + \lambda \mathcal{L}_{CDA} \tag{8}$$

## 3 Experiments

### 3.1 Datasets and experiment setup

We employed multiple EEG datasets with varying numbers of emotion categories and recording channels for model pre-training (Table 2). All datasets underwent the same preprocessing pipeline, including downsampling, filtering, ICA-based artifact removal, channel interpolation, and re-referencing. See Appendix C for details.

**SEED series.** The SEED series contains several emotional EEG datasets in which participants watch emotional video stimuli. Participants' EEG signals in response to videos were recorded with the ESI NeuroScan System (62 channels, 1000 Hz). We used SEED (15 subjects, 45 trials per subject) [7], SEED-IV (15 subjects, 72 trials per subject) [23], SEED-V (16 subjects, 45 trials per subject) [24] and SEED-VII (20 subjects, 80 trials per subject) [25] in this study.

**FACED.** The FACED dataset [9] records 32-channel EEG signals from 123 subjects as they watched 28 video clips eliciting nine emotions (amusement, inspiration, joy, tenderness, anger, fear, disgust, sadness, and neutral). The EEG signals were recorded using a wireless EEG system (NeuSen.W32, Neuracle, China) at a sampling rate of 250 or 1000 Hz. To match the number of subjects in other datasets, which allows for a fair comparison of the effect of dataset augmentation on performance, we used the first 20 subjects in this dataset.

**DEAP.** DEAP dataset [8] contains EEG recordings from 32 participants as they watched 40 one-minute music video clips. Participants reported their arousal, valence, like/dislike, dominance, and familiarity after watching each video. For each video segment, we derived binary emotion labels by performing a median split on participant ratings along the valence, arousal, and dominance dimensions.

### 3.2 Implementation details

We use a two-layer MultiLayer Perceptron (MLP) with ReLU activation and batch normalization for emotion classification. MLP inputs are features extracted from the pre-trained EEG encoder. The EEG encoder is trained using the Adam optimizer. All experiments are implemented in Python 3.12.3 using the PyTorch 2.3.1 framework and are executed on an NVIDIA GeForce RTX 3090 GPU. Details of the hyperparameters are shown in Table S1 and Table S2.

To obtain low-dimensional features relevant for emotion recognition, we average the output of the EEG encoder along the temporal dimension for each sample (with a window length of 5 seconds). The features from consecutive EEG samples within a trial are concatenated over time and smoothed using a linear dynamical system (LDS) model [7]. The smoothed features are submitted to the MLP classifier.

Table 3: Performance of cross-dataset subject-independent classification (leave-one-dataset-out evaluation).

| Dataset | Methods | Accuracy | Precision | Recall | F1 Score | AUROC |
|---|---|---|---|---|---|---|
| SEED | DE baseline | 62.35 ± 4.10 | 62.91 ± 3.76 | 62.58 ± 3.77 | 62.49 ± 3.69 | 79.38 ± 3.15 |
| | MMM | 76.24 ± 2.76 | 75.70 ± 2.59 | 75.90 ± 0.75 | 75.79 ± 1.67 | 87.73 ± 1.51 |
| | LaBraM | 71.47 ± 3.98 | 70.71± 2.78 | 71.03 ± 1.37 | 70.84 ± 1.26 | 86.15 ± 0.96 |
| | EEGPT | 71.87 ± 2.36 | 71.51 ± 4.64 | 71.99 ± 2.61 | 71.66 ± 2.35 | 85.41 ± 1.34 |
| | mdJPT (ours) | **79.65 ± 1.24** | **79.84 ± 1.17** | **79.62 ± 1.20** | **79.45 ± 1.32** | **92.98 ± 1.00** |
| SEED-IV | DE baseline | 45.44 ± 1.99 | 45.59 ± 1.96 | 45.28 ± 1.97 | 45.10 ± 1.84 | 70.03 ± 1.88 |
| | MMM | **55.61 ± 4.12** | **55.22 ± 3.94** | **55.47 ± 1.46** | **55.30 ± 2.46** | 75.87 ± 1.20 |
| | LaBraM | 47.88 ± 3.41 | 47.81 ± 2.24 | 48.01 ± 1.78 | 47.89 ± 1.29 | 69.97 ± 1.78 |
| | EEGPT | 44.10 ± 1.77 | 43.82 ± 3.56 | 44.16 ± 2.86 | 43.91 ± 2.51 | 62.32 ± 1.93 |
| | mdJPT (ours) | 53.53 ± 0.75 | 53.81 ± 1.02 | 53.67 ± 0.78 | 53.37 ± 0.79 | **77.51 ± 0.34** |
| SEED-V | DE baseline | 45.58 ± 1.92 | 46.02 ± 1.96 | 45.98 ± 1.92 | 45.55 ± 1.94 | 73.64 ± 1.54 |
| | MMM | 58.64 ± 3.56 | 58.51 ± 1.90 | 59.01 ± 3.00 | 58.68 ± 1.20 | 82.80 ± 1.94 |
| | LaBraM | 41.80 ± 3.53 | 41.69 ± 2.19 | 42.09 ± 2.71 | 41.80 ± 1.56 | 70.81 ± 2.72 |
| | EEGPT | 45.27 ± 3.66 | 44.97 ± 2.79 | 45.24 ± 2.39 | 45.06 ± 2.16 | 67.21 ± 2.06 |
| | mdJPT (ours) | **65.02 ± 0.98** | **65.06 ± 1.28** | **65.53 ± 0.66** | **64.85 ± 1.01** | **88.70 ± 0.98** |
| SEED-VII | DE baseline | 29.12 ± 1.20 | 28.24 ± 1.22 | 27.67 ± 0.90 | 27.64 ± 1.00 | 64.78 ± 0.83 |
| | MMM | 30.29 ± 2.63 | 30.32 ± 3.53 | 30.15 ± 2.23 | 30.12 ± 2.27 | 68.37 ± 1.84 |
| | LaBraM | 26.85 ± 2.57 | 26.53 ± 4.29 | 26.54 ± 2.33 | 26.45 ± 3.05 | 66.62 ± 1.87 |
| | EEGPT | 27.81 ± 2.58 | 27.78 ± 2.69 | 28.03 ± 2.89 | 27.80 ± 2.18 | 59.77 ± 1.62 |
| | mdJPT (ours) | **43.93 ± 0.69** | **43.65 ± 0.82** | **43.03 ± 0.59** | **43.07 ± 0.70** | **79.27 ± 0.44** |
| FACED | DE baseline | 19.97 ± 2.47 | 17.73 ± 1.62 | 19.16 ± 2.18 | 15.81 ± 1.53 | 60.63 ± 1.38 |
| | MMM | 22.13 ± 4.12 | 21.86 ± 5.05 | 21.88 ± 4.84 | **21.81 ± 4.74** | 59.79 ± 3.11 |
| | LaBraM | 20.35 ± 4.97 | 20.36 ± 3.79 | 20.25 ± 2.44 | 20.23 ± 2.88 | **67.59 ± 2.10** |
| | EEGPT | 21.55 ± 3.46 | 21.62 ± 3.51 | 21.62 ± 3.79 | 21.51 ± 3.31 | 59.96 ± 2.07 |
| | mdJPT (ours) | **23.46 ± 3.39** | **23.43 ± 1.13** | **22.47 ± 3.43** | 20.32 ± 3.88 | 65.53 ± 4.13 |
| DEAP | DE baseline | 55.54 ± 1.11 | 55.60 ± 1.11 | 55.54 ± 1.11 | 55.43 ± 1.13 | 56.72 ± 0.62 |
| | MMM | **72.33 ± 3.41** | 66.16 ± 7.26 | **76.80 ± 3.95** | 70.50 ± 1.70 | 77.80 ± 11.53 |
| | LaBraM | 67.45 ± 5.63 | 61.39 ± 5.58 | 65.66 ± 1.53 | 63.45 ± 3.56 | **77.83 ± 0.65** |
| | EEGPT | 63.33 ± 4.38 | 56.48 ± 6.74 | 64.73 ± 1.26 | 60.32 ± 2.77 | 74.19 ± 4.31 |
| | mdJPT (ours) | 71.71 ± 10.38 | **71.78 ± 10.42** | 71.71 ± 10.38 | **71.69 ± 10.38** | 75.75 ± 12.20 |
| Average | DE baseline | 43.00 | 42.68 | 42.70 | 48.19 | 68.13 |
| | MMM | 54.54 | 51.29 | 53.20 | 52.03 | 75.39 |
| | LaBraM | 45.97 | 44.75 | 45.60 | 45.11 | 73.16 |
| | EEGPT | 47.32 | 44.36 | 45.96 | 45.04 | 68.14 |
| | mdJPT (ours) | **56.22** | **56.26** | **56.01** | **55.46** | **79.96** |

## 3.3 Results of cross-dataset subject-independent classification

Under the cross-dataset subject-independent (few-shot) setting, our method uses a small subset of subjects from the target dataset to train an MLP classifier, which can generalize to the remaining subjects. We split the target dataset at a 1:3 subject ratio for MLP training and testing. We repeated the random split 6 times and report their mean and standard deviation. A leave-one-dataset-out cross-validation is employed to evaluate the model: the EEG encoder is pretrained on all datasets except the target set. The performance is evaluated using five metrics: accuracy, precision, recall, F1 Score, and AUROC. Our method is compared to three generic pre-trained EEG models (i.e., MMM (26), LaBraM (1), EEGPT (3)). We used the publicly available pre-trained parameters of these models. A baseline method of directly extracting DE features from EEG signals without pre-training is also implemented. Features of the comparison methods are submitted them to an LDS smoother and an MLP classifier with the same procedures as ours.

Our method obtains the best performance across all evaluation metrics on the SEED, SEED-V, and SEED-VII datasets. On SEED-IV, FACED, and DEAP, it also ranks within the top two across nearly all metrics (Table 3). In terms of average performance across all datasets, our method achieves the highest scores for all the metrics, improving the accuracy, precision, recall, F1 score, and AUROC by 1.68%, 4.97%, 2.81%, 3.43%, and 4.57% (and relative improvement of 3.08%, 9.69%, 5.28%,

6.59%, and 6.06%), respectively , over the state-of-the-art. These results demonstrate the superiority of our multi-dataset joint pre-training framework, supporting the effectiveness of task-specialized pre-training over general-purpose pre-training approaches in emotion decoding. mdJPT is also more compact than existing pre-trained models, with less trainable parameters (1.0M, Table S10). The model's performance is quite stable across different random seeds (Table S9). Further results of different classification settings on the DEAP dataset and the transfer from DEAP to other datasets are provided in Appendix E.8 and E.7, respectively. The confusion matrix of fine-grained emotion classification on the FACED dataset is provided in Appendix E.4 (Fig. S1).

## 3.4 Results of zero-shot generalization to a new dataset

Under the zero-shot setting, the EEG encoder is pre-trained on five datasets and directly evaluated on a held-out dataset without any fine-tuning. For each testing sample, we compute cosine similarities across all samples in the target dataset and identify its nearest neighbor in the representation space. A prediction is correct if the nearest neighbor shares the same emotion label. The overall accuracy reflects the encoder's zero-shot ability to produce discriminative and dataset-invariant representations. The performance of mdJPT is also compared with DE baseline, LaBraM, and EEGPT. MMM is not included as it requires fine-tuning on new datasets to optimize region-wise tokens.

Our method consistently outperforms comparison methods on all datasets by a large margin, with an improvement of 11.9% on average, and relative improvement of 40.0% over the state-of-the-art. On the SEED dataset, mdJPT outperforms the second best model, EEGPT, by 17.05%. On the challenging FACED dataset, other models performed close to chance level (11%), with only mdJPT achieving better-than-chance accuracy. On the DEAP dataset, mdJPT achieved an impressive accuracy of 73.3%, even outperforming the best fine-tuned model. This may be due to overfitting in fine-tuned models when trained on a small number of subjects with large individual differences.

Table 4: Performance of zero-shot generalization.

| Model | SEED | SEED-IV | SEED-V | SEED-VII | DEAP | FACED |
|---|---|---|---|---|---|---|
| DE baseline | 50.54 | 46.86 | 43.32 | 23.58 | 55.40 | 8.88 |
| LaBraM | 48.54 | 45.24 | 39.70 | 21.33 | 67.04 | 10.21 |
| EEGPT | 55.42 | 34.02 | 36.97 | 20.96 | 62.78 | 11.64 |
| mdJPT (Ours) | **72.47** | **50.59** | **52.91** | **33.26** | **73.34** | **17.44** |

To visualize cross-dataset alignment after joint pre-training, we compare feature distributions of DE and our mdJPT model in Fig. 3. DE features form tight dataset-specific clusters, whereas mdJPT features show improved intermixing with lower silhouette scores (Fig. S2), indicating better cross-dataset alignment. Emotion discriminability and inter-dataset consistency are analyzed in Appendices E.5 and E.6, respectively.

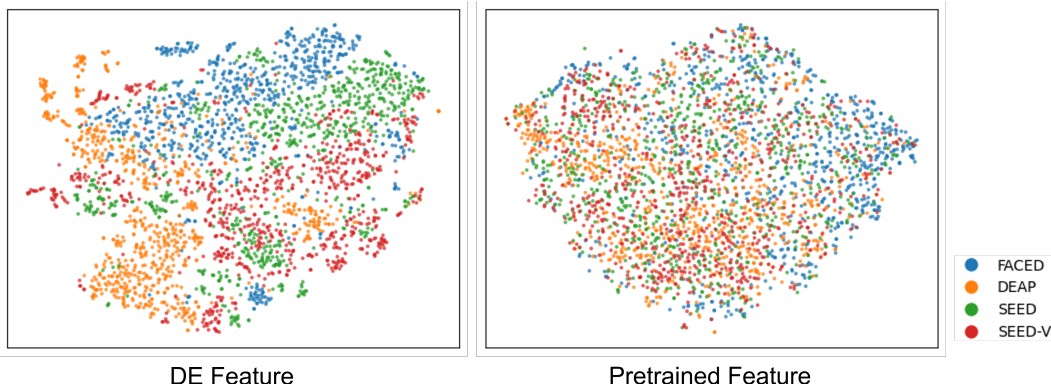

DE Feature        Pretrained Feature

Figure 3: **t-sne visualization of features**. Compared to DE, pretrained features show better cross-dataset alignment, with features from different datasets more evenly intermixed.

## 3.5 Impact of more datasets used in pre-training

To investigate the impact of using an increasing number of pre-training datasets on mdJPT's performance, we took SEED-V as the target dataset and pre-trained the model on various combinations of datasets excluding SEED-V. As shown in Fig. 4, the model's performance consistently improved as more datasets were included in the joint pre-training. Notably, in all combinations, adding a new dataset for joint pre-training always led to better results than the previous setting without it. In particular, mdJPT achieved an 8.55% improvement (15.14% relative improvement) when using the maximum number of training datasets compared to the best performance obtained with a single dataset. This highlights the advantage of joint training across multiple datasets in enhancing the model's generalization ability to unseen target data.

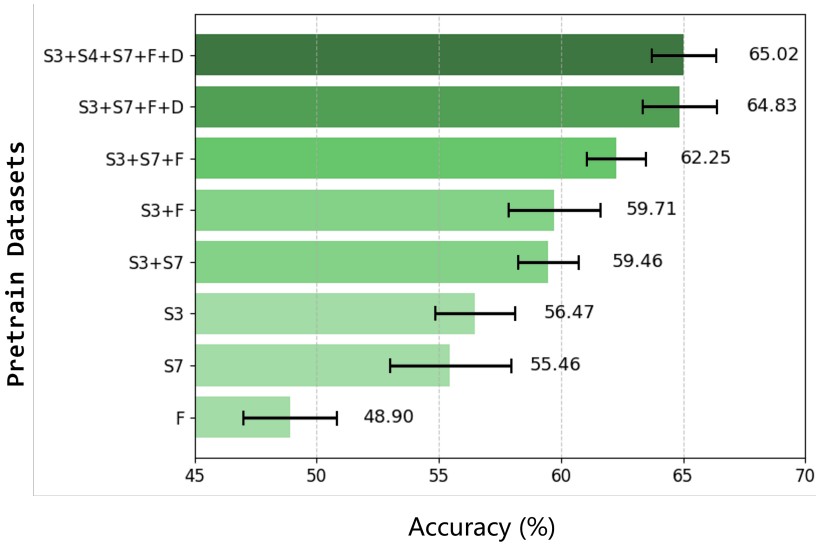

Figure 4: **Generalization performance of mdJPT with an increasing number of datasets**. Few-shot setting is used here with SEED-V dataset as the testing dataset. *Abbr.* S3, S4, S7, F, D stand for SEED, SEED-IV, SEED-VII, FACED, and DEAP, respectively.

## 3.6 Ablation Study

We conduct ablation studies with SEED-V as the target dataset to evaluate the contributions of the proposed components: CDA loss, ISA loss, and the MLLA encoder.

**CDA loss.** We use a range of CDA loss weights to examine the impact of cross-dataset covariance alignment on model performance. As shown in Table 5, introducing CDA loss with small positive factors generally improves accuracy compared to the baseline without CDA (factor = 0). The best performance is observed at a CDA factor of 0.02, achieving 65.02% accuracy. Performance slightly declines with larger factors, suggesting that overly strong alignment may hinder emotion-discriminative representation learning.

Table 5: Effect of CDA factor

| CDA factor | ACC | STD |
|---|---|---|
| 0 | 63.52 | 0.64 |
| 0.005 | 64.98 | 0.59 |
| 0.010 | 63.81 | 1.51 |
| 0.020 | 65.02 | 0.98 |
| 0.050 | 64.70 | 0.80 |
| 0.075 | 64.10 | 1.19 |
| 0.100 | 64.08 | 1.18 |

**ISA loss.** When removing the ISA loss in pre-training, the model's performance drops significantly Table (Table 6), falling even below the DE baseline. This indicates that ISA loss is not only important for aligning representations across individuals but also plays a critical role in the learning of emotion-related representations. Further comparison with supervised contrastive loss and temporally unaligned sampling strategy is provided in Appendix E.2.

Table 6: Ablation of ISA loss

| Model | Accuracy | Precision | Recall | F1 Score | AUROC |
|-------|----------|-----------|--------|----------|-------|
| DE baseline | 45.58 ± 1.92 | 46.02 ± 1.96 | 45.98 ± 1.92 | 45.55 ± 1.94 | 73.64 ± 1.54 |
| w/o $\mathcal{L}_{ISA}$ | 30.51 ± 1.16 | 24.61 ± 5.48 | 27.57 ± 1.89 | 23.21 ± 3.76 | 60.29 ± 1.95 |
| w $\mathcal{L}_{ISA}$ | **62.35 ± 4.10** | **62.91 ± 3.76** | **62.58 ± 3.77** | **62.49 ± 3.69** | **79.38 ± 3.15** |

**MLLA encoder**   To assess MLLA channel encoder's effectiveness in modeling EEG temporal dynamics, we replace it with a vanilla multi-head transformer (27). The transformer encoder uses two attention heads, treating the EEG data from all channels at each time step as a single token. It projects these tokens into a 128-dimensional latent space and then to a 32-dimensional output. The results shown in Table 7 demonstrate that the MLLA encoder consistently outperforms the vanilla Transformer across multiple metrics, confirming its advantage in capturing the temporal patterns in EEG signals. The advantage of the spatiotemporal dynamics model over a transformer layer is shown in Appendix E.1.

Table 7: Comparison of MLLA Channel Encoder with Transformer

| Model | Accuracy | Precision | Recall | F1 Score | AUROC |
|-------|----------|-----------|--------|----------|-------|
| DE baseline | 45.58 ± 1.92 | 46.02 ± 1.96 | 45.98 ± 1.92 | 45.55 ± 1.94 | 73.64 ± 1.54 |
| Transformer | 59.90 ± 1.32 | 60.39 ± 1.52 | 60.37 ± 0.75 | 59.73 ± 1.27 | **84.53 ± 1.25** |
| MLLA | **62.35 ± 4.10** | **62.91 ± 3.76** | **62.58 ± 3.77** | **62.49 ± 3.69** | 79.38 ± 3.15 |

## 4   Discussion and conclusion

In this study, we propose mdJPT, a scalable multi-dataset pretraining framework for EEG-based emotion recognition. The model supports cross-dataset transfer and demonstrates strong generalization ability, confirming that joint pretraining across datasets can significantly enhance performance on downstream tasks. To mitigate inter-dataset and inter-subject distribution shifts, we introduce a CDA loss, which aligns sample covariance across datasets and subjects. We conduct extensive experiments to validate the effectiveness of the proposed pre-training strategy. Results show that for both few-shot and zero-shot settings, the model generalizes well to target datasets and achieves superior performance to existing large-scale EEG pretraining methods, especially for the most challenging zero-shot setting. This highlights the broader applicability and flexibility of the proposed approach in real-world emotion recognition scenarios. Furthermore, this task-specific multi-dataset joint pre-training paradigm can offer valuable insights for other brain-computer interface tasks.

**Limitations and future directions.**   Our study still has several limitations. First, despite the advantage over comparison methods, the performance of mdJPT on the most challenging nine-category classification of the FACED dataset is still not satisfactory, indicating residual feature distribution shifts persist. Future work is needed to further improve the model's generalizability to more fine-grained emotions. Second, our evaluation primarily focuses on video-induced emotion paradigms. We include an exploratory analysis to test the model's generalizability to imagery-induced contexts on the EmoEEG-MC dataset (28) (Appendix E.3). Generalizability to diverse emotion elicitation paradigms requires further validation. Third, our approach addresses signal-level heterogeneity but does not model individual differences in emotional experience, potentially overlooking nuanced affective states. Future work could incorporate personalized emotion ratings through soft contrastive learning to bridge this gap. Fourth, the generalizability of our findings is constrained by the limited demographic diversity (e.g., age, culture, and health status) of current EEG emotion datasets, and the practical deployment is further limited by the cumbersome nature of current EEG hardware. The development of wearable devices may help overcome this barrier. We also note broader societal and ethical implications, including potential misuse in emotional surveillance or profiling without consent. To mitigate these risks, future efforts should establish regulatory guidelines and develop privacy-preserving frameworks for responsible model deployment.

**Acknowledgements**   The authors would like to express their gratitude to Professor Dan Zhang from the Department of Psychological and Cognitive Sciences, Tsinghua University, for his

valuable suggestions. This work was supported by the National Natural Science Foundation of China (62472206), National Key R&D Program of China (2025YFC3410000), Shenzhen Science and Technology Innovation Committee (RCYX20231211090405003, KJZD20230923115221044, RCBS20231211090748082), Guangdong Provincial Key Laboratory of Advanced Biomaterials (2022B1212010003), and the open research fund of the Guangdong Provincial Key Laboratory of Mathematical and Neural Dynamical Systems, the Center for Computational Science and Engineering at Southern University of Science and Technology.

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

# A    Related work

**Cross-dataset EEG Emotion Recognition**. Existing cross-dataset EEG emotion recognition mainly focuses on one-to-one dataset adaptation. Wang et al. (13) proposed a pre-trained vision transformer for cross-dataset EEG emotion recognition. They utilized cross-domain data mixup to blend source-target EEG distributions and adversarial domain alignment to minimize dataset-specific biases. Zhou et al. (14) proposed an Emotional EEG Style Transfer Network ($E^2$STN) for cross-dataset domain adaptation. Liu et al. (12) proposed a soft contrastive masked modeling framework with learnable weights assigned to sample pairs in contrastive learning. It achieved state-of-the-art one-to-one dataset transfer on SEED, SEED-IV, and DEAP datasets, and can be transferred to datasets with different emotion categories. These methods need to be fine-tuned on each target subject, which constrains their generalization to new subjects without training data. Imtiaz and Khan (10) proposed a target data selection method to gradually select reliable target domain samples for training. They also employed test-time augmentation when the prediction confidence is low. The model is trained on labeled source domain data and unlabelled target domain data, with evaluations performed on the target domain test set. Li et al. (11) proposed a distillation-based method, in which the knowledge from the target-domain model was distilled to the source-domain model. The subjects from the target dataset were divided into training and testing sets. The model was trained on the source dataset and training subjects of the target dataset, and tested on the left-out subject of the target dataset.

**Self-supervised Pre-training of EEG Models**. Self-supervised learning has achieved remarkable success in computer vision (CV) and natural language processing (NLP). Recently, many studies have adopted self-supervised learning frameworks to develop pre-trained models for EEG data (2). Kostas et al. proposed BENDR (29), a contrastive learning framework pre-trained on EEG recordings from over 10,000 people and evaluated on four downstream brain-computer interface tasks and a sleep-staging task. BENDR employs a convolutional encoder to extract features from local temporal windows and utilizes random masking and contrastive learning as a self-supervised pre-training strategy for EEG datasets. MMM (26) model addresses the challenges of channel discrepancies and spatial structure modeling across EEG datasets. It incorporates channel position encoding to embed 2D spatial information into representation learning. Additionally, region-level tokens are introduced to construct a hierarchical spatial representation. MMM demonstrates competitive performance in emotion recognition tasks. The Biosignal Transformer (BIOT) (5) broke the time series of channel-wise biosignals into fixed-length tokens and combined them into "sentences". Then the standard Transformer can be trained with either supervised or self-supervised learning. The Large Brain Model (LaBraM) (1) employed a vector-quantized variational auto-encoder architecture to predict the EEG spectrum by discrete neural tokens. A Transformer model was then trained to reconstruct the masked neural tokens. LaBraM used EEG data of 2500 hours and achieved state-of-the-art performance on tasks like abnormal detection and event type classification. EEGPT (3) employs a dual self-supervised learning strategy by combining the masked reconstruction loss and the alignment loss with a momentum encoder. It introduces a joint local spatial and temporal embedding strategy to learn temporal variations and spatial co-activations of EEG. EEGPT achieved state-of-the-art performance in downstream brain-computer interface tasks and sleep staging tasks with linear probing.

# B    Details of the model architecture

**The spatiotemporal dynamics model.** EEG signals exhibit complex spatiotemporal dynamics, with temporally varying spatial co-activations across the electrodes. To capture EEG dynamic properties, we firstly pass $\hat{x}$ through a trainable linear spatial projector $W_s \in \mathbb{R}^{C \times C}$, mapped to a latent space for cross-dataset covariance alignment (see section 2.3):

$$p = W_s \hat{x} \tag{9}$$

where $p \in \mathbb{R}^{C \times N_1}$. A spatial transition convolution is then employed, utilizing convolutional kernels of size $M \times L_1$ to capture spatial variation patterns across multiple time steps $L_1$. The convolutional kernels use dilations with various temporal intervals, enabling the extraction of EEG variation patterns across different time scales:

$$h^{(1)} = W_{tr} * p \tag{10}$$

where $h^{(1)} \in \mathbb{R}^{K_1 K_2 \times 1 \times N_1}$. $W_{tr} \in \mathbb{R}^{K_1 K_2 \times C \times L_1}$ is the weight in spatial-transition convolution, "$*$" denotes the convolution operation. We used group convolution with $K_2$ spatial transition

convolutional kernels for each output dimension of the MLLA transformer, resulting in $K_1 K_2$ dimensions in total. We implemented spatial transition convolution with four different dilations, with $K_1 K_2/4$ kernels for each dilation. The inputs are padded on the temporal dimension to ensure the outputs have the same temporal size $N_1$ as the inputs.

To effectively capture the evolving spatiotemporal patterns in EEG signals, we implement a local attention module to assign dynamic attention weights on latent dimensions. The module contains temporal convolution operations, dimension-wise pooling, and linear mixing to model temporal dynamics and channel interactions. The temporal convolution and dimension-wise pooling are formulated as follows:

$$A = W^{att} * h^{(1)} \tag{11}$$

$$\bar{A} = AvePool(A) \tag{12}$$

where $W^{att} \in \mathbb{R}^{K \times L_2}$ denotes learnable temporal filters. $K = K_1 K_2$ denotes the number of filters. Here, we used group convolution with one temporal filter for each dimension. Average pooling with a time step of 1 is conducted on the temporal dimension of $A$. The inputs of the temporal convolution and average pooling are padded to make sure the output $\bar{A}$ has the same size of $h^{(1)}$. Then, a linear mixing layer combines information across channels through a linear transformation:

$$h_{i\cdot}^{(2)} = \sum_{k=1}^{K} \beta_k \bar{A}_{k\cdot}, i = 1, 2, ..., K \tag{13}$$

yielding transformed features $h^{(2)} \in \mathbb{R}^{K \times N_1}$. Dynamic attention weights are then obtained through a softmax activation function:

$$h_{it}^{(att)} = \frac{e^{h_{it}^{(2)}}}{\sum_{j=1}^{K} e^{h_{jt}^{(2)}}} \tag{14}$$

producing attention weights $h^{(att)} \in \mathbb{R}^{K \times N_1}$. These dynamic weights modulate the spatiotemporal representations $h^{(1)}$ through Hadamard product:

$$h^{(3)} = h^{(att)} \odot h^{(1)} \tag{15}$$

The weighted feature representation $h^{(3)} \in \mathbb{R}^{K \times N_1}$ is ultimately employed as the output of the EEG encoder for affective state recognition.

**The projector.** A projector is employed between the output of the EEG encoder and the inter-subject alignment loss to obtain a dedicated feature representation for inter-subject alignment (ISA) loss. The projector consists of an average pooling layer and two temporal convolution layers:

$$h^{(4)} = AvePool(h^{(3)}) \tag{16}$$

$$h^{(5)} = ReLU(W^{ISA_1} * h^{(4)}) \tag{17}$$

$$h = W^{ISA_2} * h^{(5)} \tag{18}$$

where $h \in \mathbb{R}^{K \times N_1}$. $W^{ISA_1}, W^{ISA_2} \in \mathbb{R}^{K \times L_3}$ are the temporal convolution filters. Group convolution with one convolution filter for each input dimension is employed here.

## C   Data pre-processing

To ensure consistency and comparability across datasets, we implemented a standardized automatic preprocessing pipeline using Matlab. First, EEG signals were downsampled to 125 Hz and filtered with a 0.5-47 Hz bandpass filter. The signals were then segmented into trials based on the onset and offset of emotional video stimuli. For each channel, if the proportion of data exceeding a specified multiple (m) of the median value was greater than a certain percentage (n) of the trial duration, the channel was denoted as noisy. We used two sets of thresholds: m=3, n=0.4 to identify long-lasting artifacts, and m=30, n=0.01 to detect short-term large artifacts. Noisy channels identified in this manner were interpolated using their three nearest neighboring channels. Next, independent component analysis (ICA) was applied to remove artifacts caused by eye movements or muscle activity. We utilized automatic component labeling tools in EEGLab. Components labeled as eye-

or muscle-related with a confidence level exceeding 0.8, the default thresholds in EEGLab, were removed. During the initial noisy channel detection, we observed that frontal channels (such as Fp1 and Fp2) were frequently identified as noisy, which interfered with the subsequent ICA-based detection of eye movement components. To address this issue, we excluded channels Fp1, Fp2, F7, and F8 from the initial noisy channel interpolation. After applying ICA, the noisy channel detection procedure was conducted again, including all channels at this time. Finally, the data were re-referenced to the common average.

## D  Hyperparameter Settings

The hyperparameters in model pre-training and fine-tuning are shown in Table S1. The hyperparameters in the EEG encoder, including the MLLA channel encoder and the spatiotemporal dynamics model are shown in Table S2.

Table S1: Hyperparameters of pre-training and fine-tuning.

|  | Hyperparameters | Values |
| --- | --- | --- |
| **Pre-training** | epochs | 20 |
|  | learning rate | 0.0005 |
|  | weight decay | 0.0001 |
|  | ISA loss temperature | 0.07 |
|  | window length | 5 seconds |
|  | stride | 2 seconds |
|  | Weight of CDA loss | 0.02 |
| **Fine-tuning** | batch size | 256 |
|  | learning rate | 0.0005 |
|  | weight decay | 0.0022 |
|  | hidden units | 128 |
|  | epochs | 25 |

Table S2: Hyperparameters of the EEG encoder.

|  | Hyperparameters | Values |
| --- | --- | --- |
| **MLLA channel encoder** | patch size | 32 |
|  | patch stride | 6 |
|  | hidden dim | 128 |
|  | out dim | 32 |
|  | depth | 2 |
|  | attention head number | 8 |
| **Spatiotemporal dynamics model** | number of spatial-transition convolution | 4 |
|  | time length of spatial-transition convolution | 3 |
|  | dilations | [1,3,6,12] |
|  | length of temporal filters | 15 |
|  | length of average pooling | 15 |
| **Projector** | length of temporal filters | 3 |
|  | temperature of Softmax | 1.0 |

## E  Additional results

### E.1  The effectiveness of spatiotemporal dynamics model

To assess the effectiveness of the spatiotemporal dynamics model in modeling EEG temporal dynamics, we replace it with a Transformer layer comprising multi-head self-attention (27). The

Transformer's attention mechanism considers global interactions of all temporal positions to capture dependencies. Table S3 demonstrates that the spatiotemporal dynamics model outperforms the Transformer across all metrics, with an increase of 1.39% in accuracy, confirming its advantage in capturing the spatiotemporal dynamic patterns in EEG signals.

Table S3: Comparison of the spatiotemporal dynamics model with Transformer

| Model | Accuracy | Precision | Recall | F1 Score | AUROC |
|---|---|---|---|---|---|
| DE baseline | 45.58 ± 1.92 | 46.02 ± 1.96 | 45.98 ± 1.92 | 45.55 ± 1.94 | 73.64 ± 1.54 |
| Transformer | 63.63 ± 0.96 | 64.13 ± 1.63 | 63.56 ± 1.35 | 63.35 ± 1.09 | 87.70 ± 0.44 |
| Spatiotemporal dynamics | **65.02 ± 0.98** | **65.06 ± 1.28** | **65.53 ± 0.66** | **64.85 ± 1.01** | **88.70 ± 0.98** |

## E.2 Comparision of contrastive loss and sampling strategy

To demonstrate the effectiveness of the ISA loss over other contrastive loss, we compared it with supervised contrastive learning (supervised CL), which regards samples with the same emotion labels as positive pairs and those with different emotion labels as negative pairs in each dataset. We use SupCon loss (30) and keep all other settings the same. Our strategy outperforms supervised contrastive learning by a large margin (Table S4), indicating the effectiveness of our ISA loss for EEG alignment.

We also evaluated the superiority of the proposed sampling strategy to alternatives. In the proposed temporally aligned sampling strategy, two samples in a positive pair come from the same trial and start at the same timestamp. In other words, participants were watching the same video scenes in a positive pair. To test the necessity of temporal alignment of positive pairs, we draw positive samples from the same trial but with randomly selected unmatched start timestamps. Without temporal alignment, the model performance drops significantly (Table S5). This could be due to that the temporally aligned samples provide an "anchor" for mitigating the large inter-individual differences and extracting meaningful shared representations across subjects.

Table S4: Comparison of self-supervised frameworks

| Method | Accuracy | Precision | Recall | F1 Score | AUROC |
|---|---|---|---|---|---|
| Supervised CL | 47.99 ± 2.51 | 47.89 ± 2.22 | 47.86 ± 2.41 | 47.53 ± 2.33 | 76.28 ± 1.73 |
| ISA (ours) | **65.02 ± 0.98** | **65.06 ± 1.28** | **65.53 ± 0.66** | **64.85 ± 1.01** | **88.70 ± 0.98** |

Table S5: Comparison of alignment method

| Method | Accuracy | Precision | Recall | F1 Score | AUROC |
|---|---|---|---|---|---|
| Not aligned | 55.47 ± 1.71 | 55.17 ± 1.41 | 55.56 ± 1.70 | 54.94 ± 1.49 | 80.97 ± 1.29 |
| Aligned (ours) | **65.02 ± 0.98** | **65.06 ± 1.28** | **65.53 ± 0.66** | **64.85 ± 1.01** | **88.70 ± 0.98** |

## E.3 Generalization to the imagery context

Most widely used EEG emotion datasets employ the video-induced paradigm. To assess the model's generalizability beyond video stimuli, we further perform experiments on the EmoEEG-MC dataset (28), a multi-context dataset including imagery-induced paradigm (guided narratives with active imagination), eliciting more internally driven and sustained emotions. We evaluate on the imagery task using the few-shot setting: pre-train on six video-induced datasets, then fine-tune on 1/4 of EmoEEG-MC subjects and test on the remaining. Our method outperforms the DE baseline in the imagery-induced context (Table S6), indicating the effectiveness of pre-training in context-independent representation learning. It should be noted that the performance of transfer to imagery context remains relatively low. Future work is needed to mitigate the cross-context discrepancy.

Table S6: Accuracy on EmoEEG-MC dataset with imagery-induced emotion

| Model | Accuracy | Precision | Recall | F1 Score | AUROC |
|---|---|---|---|---|---|
| DE baseline | $18.38 \pm 1.12$ | $17.91 \pm 1.62$ | $18.38 \pm 1.12$ | $17.87 \pm 1.37$ | $54.11 \pm 1.36$ |
| mdJPT (ours) | $\mathbf{20.56 \pm 0.79}$ | $\mathbf{20.83 \pm 0.66}$ | $\mathbf{20.56 \pm 0.79}$ | $\mathbf{19.85 \pm 2.11}$ | $\mathbf{56.14 \pm 0.78}$ |

### E.4 Confusion matrices

To investigate the detailed confusion patterns on FACED and EmoEEG-imagery, we calculated the confusion matrices on these datasets (Fig. S1). Each row represents a true label, and each column represents a predicted label. On the FACED dataset, the confusion matrix shows a diagonal dominance: within each row, the diagonal entry is the highest, meaning the model is more likely to predict the true label than any other single label. Among all categories, disgust, tenderness, and neutral have the highest classification accuracy, indicating they have more discriminative neural features. Joy tends to be confused with amusement or inspiration, indicating more similar neural representations across these fine-grained positive emotion categories. On the EmoEEG-imagery dataset, neutral and tenderness have the best recognition accuracy, indicating the cross-context neural similarity for these emotions.

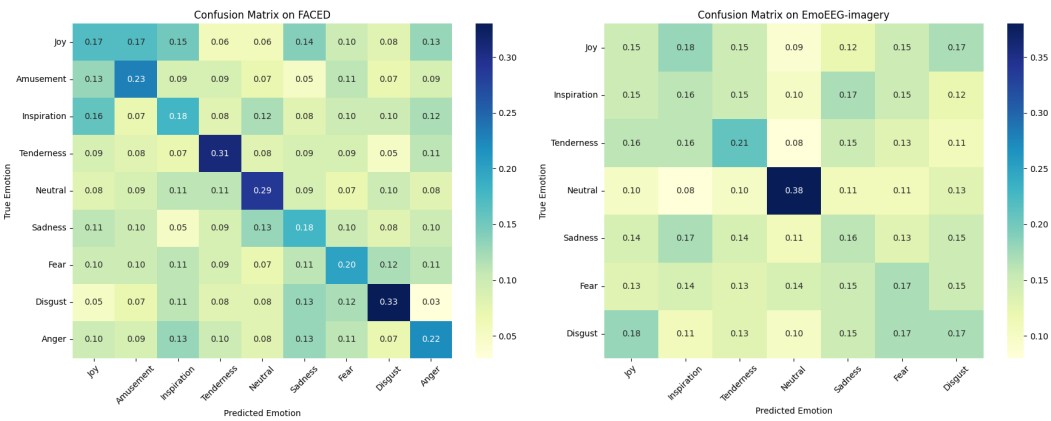

Figure S1: Confusion matrices of emotion classification results on FACED and EmoEEG-imagery datasets.

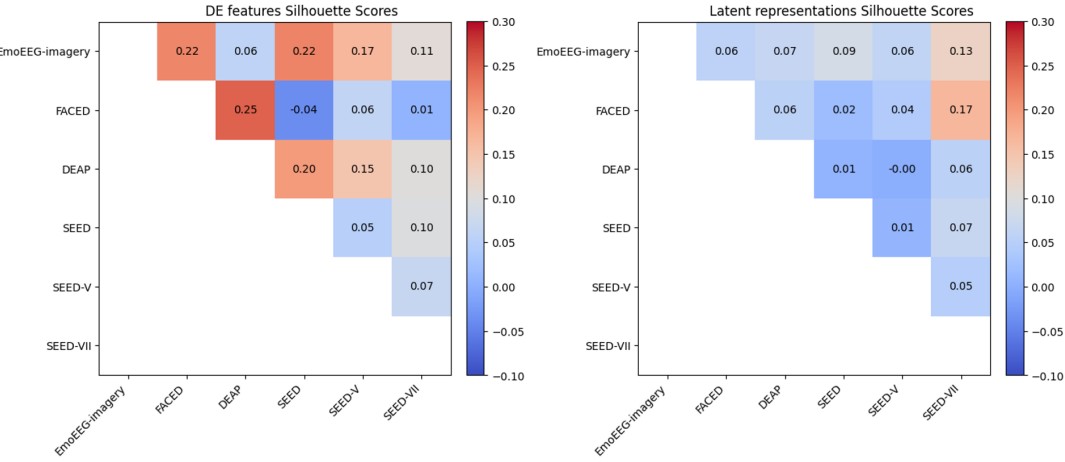

Figure S2: Average silhouette score between features of different datasets.

### E.5  Feature visualization

We quantize the clustering effect between any two datasets before and after pre-training. We regard samples from one dataset as a cluster and calculated the silhouette score of different clusters. The silhouette score decreases for most dataset pairs, indicating a more blended representation across datasets (Fig. S2).

To illustrate the effect of joint pre-training on emotion discriminability of the data representations, we visualized the distribution of EEG features on each dataset using the t-SNE method (Fig. S3). We compared EEG representations extracted by mdJPT and the DE features. Different colors represent different emotion categories. Although emotion labels are not used in pre-training, features extracted by mdJPT exhibit a higher degree of clustering within the same emotion category. This demonstrates the effectiveness of self-supervised learning in facilitating downstream emotion classification tasks. Notably, the model used for feature extraction on a specific dataset is pre-trained on all other datasets, which never accesses data from the target dataset.

### E.6  Inter-dataset consistency of emotion-related features

To validate whether our joint multi-dataset pre-training approach successfully learns emotion representations with cross-dataset invariance, we conducted comparative analyses of feature importance in emotion classification across different datasets. Specifically, we employed the Integrated Gradients method to quantify the contribution of EEG encoder output features to emotion classification decisions. This interpretability technique enables the identification of critical feature dimensions underlying the classification of each emotion category. Next, we compared the similarity of importance attributions across different datasets to identify whether the same emotion category across different datasets shares similar importance attributions. Fig. S4 illustrates the importance attributions of neutral, negative/sad, and positive/happy emotions on SEED, SEED-IV, SEED-V, and SEED-VII. We sorted the attributions of each emotion in the SEED dataset in descending order. The resulting sorted indices were then applied to reorder the attributions of the corresponding emotion categories in other datasets. As shown in the figure, similar attribution patterns can be observed across the same emotion categories in different datasets.

To more clearly illustrate this correlation, we computed the Pearson correlation of emotion attributions for any emotion pairs across datasets (Fig. S5). The attributes of identical or similar emotions across different datasets exhibit stronger correlations than different emotions. For example, feature attributes of the positive emotion on the SEED dataset exhibit a high correlation with those of the happy emotion on the SEED-IV datsaet, and attributes of the negative emotion on the SEED dataset have a high correlation with those of the sad and fear emotions on the SEED-IV dataset.

### E.7  Transfer from DEAP to other datasets

To further test whether the model can effectively transfer across datasets with different emotion categorization paradigms, we trained our model on DEAP and transferred it to SEED-series and FACED datasets. DEAP dataset employs a dimensional emotion characterization and other datasets employ discrete emotion characterization. We found that transfer from DEAP to other dataset yielded a comparable performance to that of transfer from SEED dataset (Table S7). The performance is only slgntly lower than from SEED on SEED series datasets, and is even higher on FACED dataset. This indicates the model's capability in generalization to new emotion categorization paradigms.

Table S7: Cross-dataset transfer performance between DEAP and SEED-series datasets

| Accuracy | from DEAP | from SEED |
|---|---|---|
| to SEED | $66.77 \pm 3.33$ | / |
| to SEED-IV | $46.24 \pm 0.87$ | $48.52 \pm 1.65$ |
| to SEED-V | $54.55 \pm 2.13$ | $55.85 \pm 2.31$ |
| to SEED-VII | $36.35 \pm 1.04$ | $37.33 \pm 1.19$ |
| to FACED | $22.83 \pm 1.48$ | $18.95 \pm 1.79$ |

## E.8    Other classification settings on the DEAP dataset

To comprehensively evaluate the model on dimensional emotion representation, we conducted binary classifications of high/low arousal and high/low dominance, as well as a four-quadrant valence–arousal classification on the DEAP dataset. As shown in Table S8, the model substantially outperformed the baseline across all tasks, demonstrating strong generalization to dimensional emotion representations and adaptability to different affective dimensions.

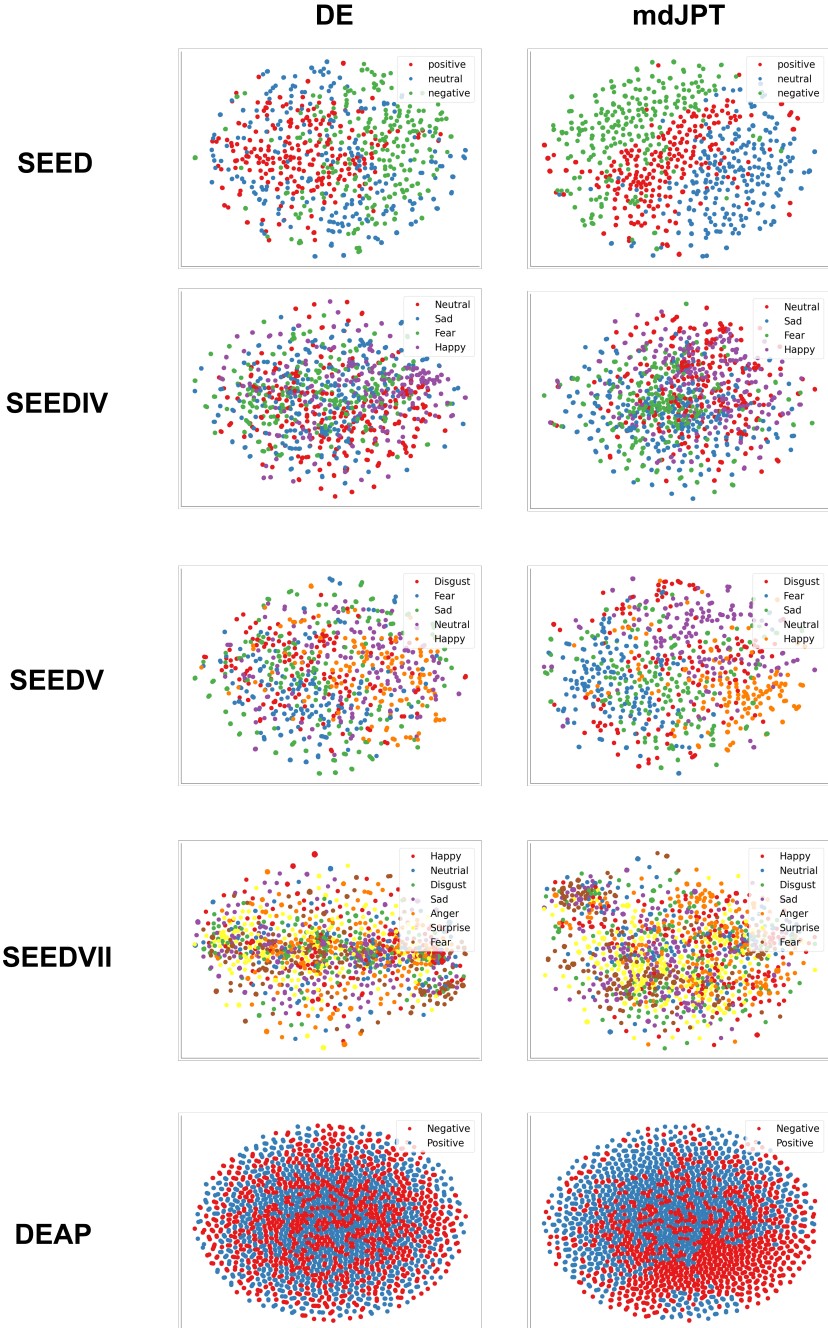

Figure S3: t-SNE visualization of extracted feature on SEED, SEED-IV, SEED-V, SEED-VII, and DEAP datasets. The EEG encoder used for feature extraction was trained on all datasets other than the target dataset.

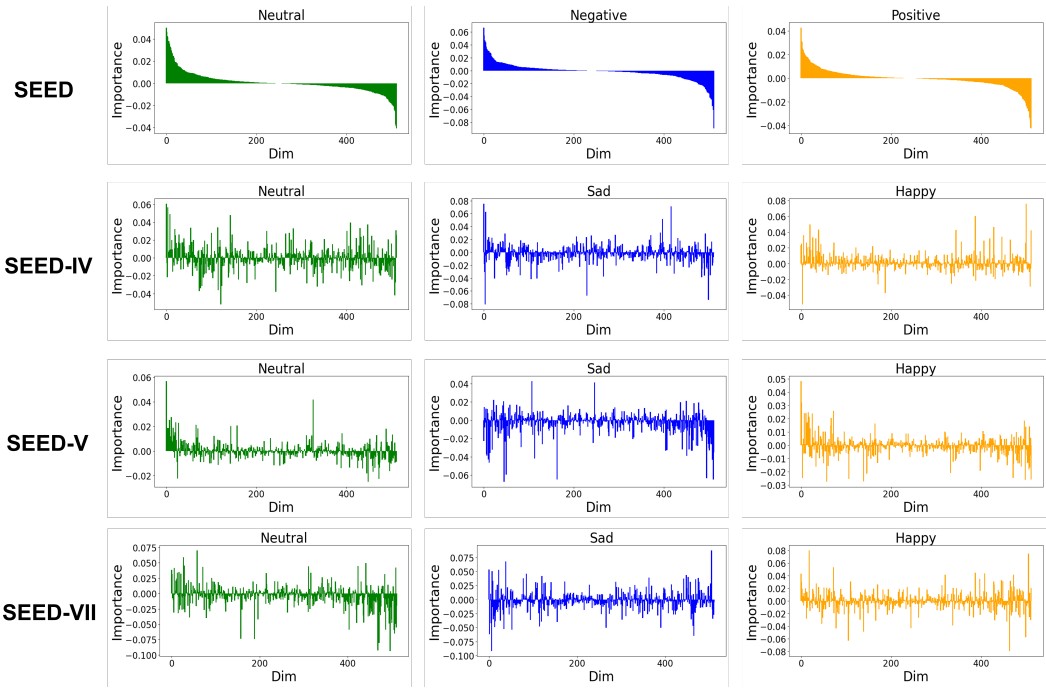

Figure S4: Comparison of feature importance attribution for neutral, negative/sad, and positive/happy emotions on SEED, SEED-IV, SEED-V, and SEED-VII datasets. The feature indices are sorted by the feature importance on the SEED dataset.

### E.9 Results of varying random seeds

We replicated the full pipeline five times with different random seeds. The performance is quite stable across the five repetitions, with a variation of less than 0.85% (Table S9). The experiment is conducted on the SEED-V dataset. This result demonstrates the stability of mdJPT training.

Table S8: Other classification settings on the DEAP dataset

| Evaluation | Method | Accuracy | Precision | Recall | F1 Score | AUROC |
|---|---|---|---|---|---|---|
| Arousal | DE baseline | 58.67±1.27 | 59.00±1.60 | 58.67±1.27 | 58.36±1.27 | 60.67±1.97 |
| | mdJPT | **69.36±1.54** | **69.60±1.68** | **69.36±1.54** | **69.27±1.52** | **74.19±0.92** |
| Dominance | DE baseline | 55.01 ± 1.17 | 55.29±1.47 | 55.01±1.17 | 54.56±0.92 | 56.08±1.89 |
| | mdJPT | **73.52±1.61** | **73.64±1.47** | **73.52±1.61** | **73.48±1.67** | **79.47±1.50** |
| Valence-arousal | DE baseline | 34.03±1.06 | 33.21±2.84 | 32.59±1.21 | 30.04±1.38 | 57.18±1.02 |
| | mdJPT | **52.94±1.42** | **50.47±1.23** | **51.10±1.83** | **49.68±2.90** | **75.78±1.34** |

Table S9: SEED-V result with varying random seeds

| Random seed | Accuracy |
|---|---|
| 0 | 65.14 ± 0.97 |
| 1 | 65.87 ± 1.08 |
| 2 | 65.74 ± 1.50 |
| 3 | 65.37 ± 1.52 |
| 19260832 | 65.02 ± 0.98 |

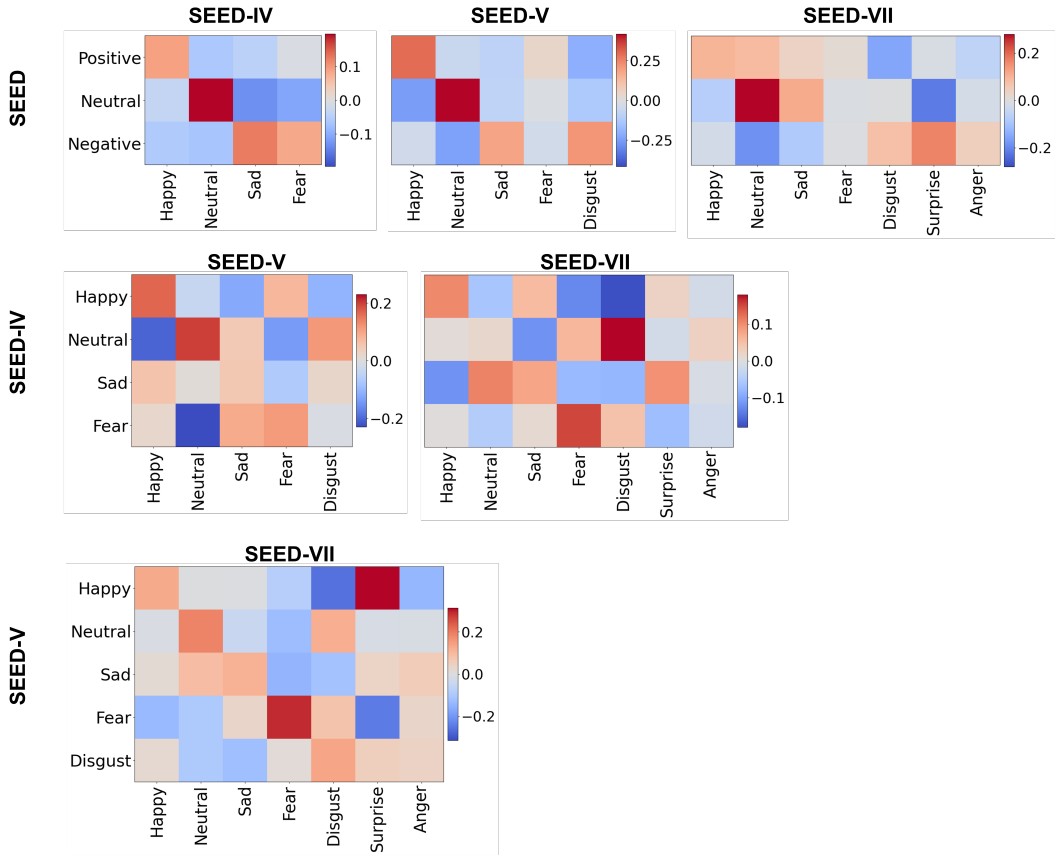

Figure S5: Pearson correlation coefficient between emotion-related feature importance weights on different datasets

## E.10 Model size comparison

To demonstrate the efficiency of our model, we compared the number of trainable parameters with several existing methods. As shown in Table S10, our model has fewer parameters of 1.0M but still outperforms state-of-the-art models, indicating a more compact and efficient design.

Table S10: Comparison of parameter size across pre-trained models.

| Model | Parameters |
|---|---|
| MMM | 1.2M |
| LaBraM | 5.8M |
| EEGPT | 4.7M |
| mdJPT (Ours) | **1.0M** |

