# OpenReview forum: "Multi-dataset Joint Pre-training of Emotional EEG Enables Generalizable Affective Computing"
_NeurIPS.cc/2025/Conference — NeurIPS 2025 poster_

### Official Review · Reviewer_YfAh · 2025-06-09

**Clarity:** 3
**Significance:** 4
**Originality:** 3
**Rating:** 4
**Confidence:** 5

**Summary:**

This work presents a framework called mdJPT for EEG-based emotion recognition that can pre-train on multiple datasets by aligning second-order statistical properties. The framework contains three key components: (1) mamba-like linear attention based EEG patchifier and spatialtemporal dynamics model, (2) cross-dataset alignment loss, (3) inter-subject alignment loss. This papers' design can be seperated into two parts: (1) EEG encoder, which has MLLA patchifier and spatialtemporal modeling, (2) domain/representation alignemtn, which has CDA and ISA. The framework is compared with three EEG foundation models on 6 datasets. Notably, it is not a traditional self-supervised framework, but requires data labeled with trials ID, each trial means a single video stimulus.

**Questions:**

1. In Figure 2. C, the color of these arrows are somewhat close and difficult to differenciate.

Typo:
1. line 48: there is an 1 outside the brackets.

**Ethical Concerns:**

["NO or VERY MINOR ethics concerns only"]

**Final Justification:**

This paper presents an interesting framework for trial-based EEG emotion recognition task.

**Limitations:**

Yes, the authors have adequately addressed the limitations and potential negative societal impact of their work.

**Quality:**

2

**Strengths And Weaknesses:**

## Strength

1. Training on multiple datasets for a specific task is a promising approach in the EEG field, I appreciate this idea.

2. Validating the method on 6 datasets is a hard work, and I appreciate your effort in trying various combination of pre-training datasets in Fig 3.

3. Opensourcing the code is very good in EEG-based emotion recognition.

## Weaknesses
1. The problem you are trying to solve is absolutely meaningful, however, your method seems to be less motivated. You used many techniques in mdJPT, but what's the rational behind these designs? For instance, why do you use mamba-like linear attention, just because it's better than the vanilla Transformer in Table 4? Aligning the covariance of two domain is also not a new idea, but why do you use it, just because a previous work [16]? Have you tried other domain alignment strategies, like mmd loss, or other statistical properties? Please note that I am not asking for more ablation study, but explanation behind all designs. Acutally, I think there is no need to design the EEG encoder for a framework paper, e.g., adding an MLLA encoder is somewhat abrupt and distracts the research focus of the article.

2. The baseline models are self-supervised framework which requires no label of data, while mdJPT requires more information about which trial does each EEG segment belong to. Although I appreciate the design to use trial information as it is a task-specific design, it brings some inconvenience and unfairness when comparing with baselines. Thus, I think it is better to discuss more about this design, rather than MLLA encoder, like comparing with more variants with mdJPT, or with some adpated baselines that can use trial information for pre-training.

3. The design of the data selection for each training batch is interesting, but you didn't discuss about it. I think this part is the most essential for ISA, as the contrastive learning is very relevant to the training batch. So it might be more appropriate to move the part of how to select a training batch (line 118-121) to section 2.4. And it is important to discuss the design of training batch selection with more experiments.

4. The types of emotions studied in different datasets are different but overlap, like SEED's all emotion types are contained in SEED-IV to SEED-VII, but some datasets like DEAP, which uses valence-arousal model to define emotion, is quite different from SEED series, which uses the discrete emotion model. How to transfer the knowledge of DEAP to SEED is an interesting problem too. But in Figure 3, this problem is overlooked. Using trial ID is an interesting approach, but in DEAP, different subjects may give different valence / arousal value to the same stimulus, how did you tackle this question? This is very important, because the design of mdJPT seems to assume each subject will have the same emotion when viewing the same video.

In conclusion, while I appreciate the framework, the vague motivation behinds the design and experiments that neglect key components make this version is not ready for publication. I hope you can refine it for the next conference / journal.

---

> ### Author Rebuttal · Authors · 2025-07-31
>
> We sincerely thank you for your insightful suggestions. In the revision, we clarify the design rationale of EEG covariance alignment and dynamics modelling. We add experiments of supervised contrastive learning, batch construction methods (temporal alignment vs. random) and batch-size analyses. Besides, we show DEAP to other datasets transfer obtained comparable performance with SEED to others, supporting cross-rating-paradigm generalization capability of our model.
>
> ---
>
> ### Q1. The motivation behind the model designs.
>
> Our work aims to develop a unified framework for multi-dataset EEG training and cross-dataset generalization. All components of our design are centered around this objective. Specifically, we introduce two complementary alignment losses: the Covariance Distribution Alignment (CDA) loss and the Inter-Subject Alignment (ISA) loss. The CDA loss aligns global second-order statistics (i.e., inter-channel covariance structures) across subjects and datasets. This captures **inter-region coupling** (or functional connectivity) patterns in EEG—features that have been increasingly emphasized in recent neuroscience and EEG decoding literature [1-3]. While MMD with a Gaussian kernel can capture higher-order statistics, there is currently limited evidence in the EEG literature suggesting the functional relevance of such statistics. Moreover, MMD introduces sensitivity to kernel bandwidth and other hyperparameters. Therefore, we prioritize CDA loss as a **simple, interpretable, and stable** alignment objective in our framework.
>
> For the model architecture, we aim to design a **physiologically plausible and computationally efficient** EEG encoder. We were not fully satisfied with standard Transformer architectures due to their limited interpretability and relatively large parameter counts. As such, we propose a tailored architecture consisting of an MLLA-based Channel Encoder and a Spatiotemporal Dynamics Module. The former captures rich temporal dynamics from each EEG channel, while the latter integrates information across channels and time, enabling the model to learn coordinated patterns of brain activity.
>
> The MLLA component is particularly suited for modeling **long-range dependencies** in EEG due to its combination of an input gate, linear attention, and a forget gate. We apply overlapping strided patches (see Fig. 2D), where each patch is first linearly projected—functionally similar to strided convolution—to extract **local features**. These are then processed by linear attention for **global dependency** modeling, followed by a forget gate to **filter salient patterns**. In contrast, standard Transformers use the QKᵀV mechanism to assign weights to input tokens by computing dot products between queries and keys, which, in our view, lacks physiological priors to effectively capture EEG-specific characteristics. The Spatiotemporal Dynamics Module further captures inter-channel dynamics and uses local attention to emphasize important spatial-temporal patterns.
>
> While model architecture is not the main contribution of this work, we designed our encoder based on domain-specific understanding of EEG data and validated its effectiveness via lesion studies. We believe this principled design—though not exhaustively benchmarked against all alternatives—should not undermine the merit of our proposed framework. In the revised version, we will further explain the motivation of our framework design and discuss its relashionship with other methods.
>
> [1] Zhang Y, Chen Z S. Harnessing electroencephalography connectomes for cognitive and clinical neuroscience[J]. Nature Biomedical Engineering, 2025: 1-16.
> [2] Li C, Tang T, Pan Y, et al. An efficient graph learning system for emotion recognition inspired by the cognitive prior graph of eeg brain network[J]. IEEE Transactions on Neural Networks and Learning Systems, 2024, 36(4): 7130-7144.
> [3] Wang Y, Qiu S, Ma X, et al. A prototype-based SPD matrix network for domain adaptation EEG emotion recognition[J]. Pattern Recognition, 2021, 110: 107626.
>
> ---
>
> ### Q2. The comparison with other self-supervised frameworks using trial information.
>
> To demonstrate the effectiveness of our self-supervised strategy over others using trial information, we compared it with **Supervised contrastive learning (supervised CL)**, which regards samples with the same emotion labels as positive pairs and those with different emotion labels as negative pairs in each dataset. It directly optimizes based on emotion labels (but agnostic to specific emotion categories). We use SupCon loss [4] and keep all other settings the same. Our strategy outperforms supervised contrastive learning by a large margin, indicating the effectiveness of our ISA loss for EEG alignment.
>
> | Method        | Accuracy | Precision | Recall | F1 Score | AUROC |
> | ------------- | -------- | --------- | ------ | -------- | ----- |
> | Supervised CL |47.99±2.51|47.89±2.22|47.86±2.41|47.53±2.33|76.28±1.73|
> | Ours          |65.02±0.98|65.06±1.28|65.53±0.66|64.85±1.01|88.70±0.98|
>
> [4] Khosla P, Teterwak P, Wang C, et al. Supervised contrastive learning[J]. Advances in neural information processing systems, 2020, 33: 18661-18673.
>
> ---
>
> ### Q3. The effects of training batch selection in pre-training.
>
> Indeed, the trianing batch selection is carefully designed to learn subject-invariant representations. Sampling two temporally aligned samples as the positive pair is critical. In the preliminary experiment, we found that when the two positive samples are randomly selected from the same trial without temporal alignment, the model has a near random top1 accuracy in contrastive learning. This is possibly due to that temporally aligned samples carry more shared information, which facilitates the model to focus on meaningful shared representations across subjects and discard subject-specific noise. We will add these comparison experiments in the revised version.
>
> We also experiment with larger batch sizes in pre-training. As previous work in contrastive learning, such as SimCLR, demonstrated that contrastive learning benefits from a larger batch size, we increase our batch size to double: Obtain two samples from each trial in one batch (one sample from each trial was obtained in the previous version). Samples that are not temporally aligned across subjects are regarded as negative samples.
>
> We found doubling the batch size increases the leave-one-dataset-out few-subject fune-tuning accuracy on the SEED-V dataset by 2.08% and by 2.27%. This demonstrates that larger batch size also benefit EEG contrastive learning for learning effective representations. However, doubling the batch size also increases the occupation of GPU memory from approximately 20G to 40G in joint pre-training, which poses a higher demand of the hardware. Future work is needed to more comprehensively investigate the effect of batch size on downstream performance.
>
> ---
>
> ### Q4.1 Transfer from the DEAP dataset to SEED datasets.
>
> We conducted experiments to train the model on the DEAP dataset and transfer it to other datasets. In the table, we compare the performance of transfer from DEAP with transfer from SEED.
>
> | Accuracy        | from DEAP | from SEED |
> | ---------------- | -------- | --------- |
> | to SEED     |66.77±3.33|/|
> | to SEED-IV  |46.24±0.87|48.52±1.65|
> | to SEED-V   |54.55±2.13|55.85±2.31|
> | to SEED-VII |36.35±1.04|37.33±1.19|
> | to FACED |22.83±1.48|18.95±1.79|
>
> The performance of transfer from DEAP is slightly lower than transfer from SEED to other SEED-series datasets, but the performance gap is within 2.3%. Besides, the accuracy of transfer from DEAP to FACED is higher than transfer from SEED to FACED by 3.9%. These results demonstrate the effectiveness of our framework in transferring between datasets with discrepancies in emotion label paradigms. It aligns with our expectation that the model can learn generalizable emotion representations through contrastive learning, which do not rely on emotion label information.
>
> ---
>
> ### Q4.2 The issue of individual differences in valence/arousal experience on the DEAP dataset.
>
> We agree that inter-subject variability is a fundamental challenge in affective EEG analysis. Such variability arises from two sources:
> (1) Differences in emotional perception (i.e., subjects experiencing different emotions in response to the same stimulus)
> (2) Differences in neural signal properties that are unrelated to emotion (e.g., due to montage, scalp geometry, or baseline spectral patterns)
>
> In this work, we focus on mitigating the second type—signal-level inter-subject variability, which poses a major obstacle to generalization across subjects and datasets. Modeling both types of variability simultaneously would significantly complicate the training process and potentially reduce robustness.
>
> To address this, mdJPT leverages temporally aligned EEG segments as a weak prior for capturing shared neural patterns. Although such alignment does not ensure identical emotional states across subjects, it increases the likelihood of emotion similarity compared to unaligned segments. This provides a practical and effective anchor for extracting robust emotion-related representations.
>
> We acknowledge the importance of accounting for subjective emotional differences, and we plan to incorporate more individualized modeling strategies (e.g., soft contrastive labels based on individualized emotion ratings) in future work.

---

> > ### Comment · Reviewer_YfAh · 2025-08-03
> > **Response to the authors**
> >
> > I appreciate the authors' efforts. I would be glad if you address this limitations:
> >
> > According to your answer to Q.4, you process DEAP just the same way as SEED. Then, the figure 2. C is not that appropriate, as you assume each subject has the same emotion when viewing the same video.
> >
> > Moreover, please adding Q4.2 in the limitation part of future version.
> >
> > Plus, you provided some further results in Q2 and another comment, pls clarify the details of these experiments, I do not know the datasets they used. And what is strictly temporally aligned means?

---

> > > ### Author Response · Authors · 2025-08-04
> > > **Limitations explanation and methodology clarification**
> > >
> > > Thank you for your timely feedback. We will revise Fig. 2 and add a discussion of the limitations. We clarify the experimental details and sampling strategy below.
> > >
> > > **(1) The revision of Fig. 2C.** We agree that Fig. 2C is potentially misleading—it may suggest that positive sample pairs necessitate matching discrete emotion experience labels, whereas our methodology only requires identical stimuli. We will substitute the emotion emojis with illustrations of different stimuli to demonstrate that positive samples correspond to subjects watching the same video, but not necessarily experiencing the same emotion.
> > >
> > > **(2) The limitation of overlooking individual difference in emotion experience.** We will add the following limitation in the *Limitations and future directions* section:
> > >
> > > >"Our work mainly focuses on mitigating signal-level inter-subject and inter-dataset variability, but does not consider the inter-subject difference of emotional experience. This simplification, though practical for generalization, may overlook nuanced individualized affective states. Future work can incorporate individualized modeling strategies, such as soft contrastive learning based on personalized emotion ratings, to address this gap."
> > >
> > > **(3.1) The clarification of experimental details in the response.** In the response, experiments used SEED-V as the target dataset and the other datasets as training datasets. For supervised contrastive learning, only the loss was substituted by the SupCon loss, which regards samples with the same emotion labels as positive pairs. Other experimental settings are exactly the same as the main experiments in Table 3 of the original manuscript. For the training batch selection experiment, only the sampling strategy was changed as *not aligned* (its meaning will be detailed in the following paragraph), with other settings kept unchanged.
> > >
> > > **(3.2) Clarification of the sampling strategy (strictly temporally aligned).**  In the proposed temporally aligned sampling strategy, two samples in a positive pair come from the same trial and *start at the same timestamp*, which we denoted as "receiving the same emotional stimulus" in the original manuscript. In other words, participants were watching the same video scenes in a positive pair. Samples have a time length of 5 seconds (as denoted in Appendix E). *Not aligned* means that a positive pair comes from the same trial but with randomly selected unmatched start timestamps. We assume that the temporally aligned samples provide an "anchor" for mitigating the large inter-individual differences and extracting meaningful shared representations across subjects. We will clarify the *temporally aligned* sampling strategy more explicitly in the revised version.
> > >
> > > Thank you again for raising these questions and helping us enhance the paper's clarity.

---

> > > > ### Comment · Reviewer_YfAh · 2025-08-05
> > > > **Response to the authors**
> > > >
> > > > Thanks for your answers. I decide to raise my rating to 4.

---

> > > > > ### Author Response · Authors · 2025-08-07
> > > > >
> > > > > We sincerely thank you for your recognition of our work. During the discussion session, we further completed the zero-shot experiment to test the model's generalization ability to unseen datasets without fine-tuning. Our model outperforms other EEG foundation models on three representative datasets. We compute cosine similarity across samples and define zero‑shot accuracy by whether the top‑similar pair shares the same label. Samples from the same trial are excluded to avoid leakage. Here, MMM is not included because it necessitates fine-tuning on a new dataset to optimize the region-wise tokens. To our knowledge, this is the first zero‑shot cross‑dataset EEG emotion experiment. These results highlight the generalizability of our framework.
> > > > >
> > > > > **Table: Comparison of different models under zero-shot settings.**
> > > > >
> > > > > |Top1 accuracy|  SEED-V| FACED | DEAP |
> > > > > | ----------- | -------|-------|------|
> > > > > | DE baseline | 43.3  |8.9 |55.4|
> > > > > | LaBram      | 39.7  |10.2|67.0|
> > > > > | EEGPT       | 37.0  |11.6|62.8|
> > > > > | mdJPT       |**52.9**|**17.4**|**73.3**|
> > > > >
> > > > > Besides, we conducted further analyses to understand the data discrepancy across datasets before and after pre-training, the confusion matrices of emotion classification, and the stability of the methods over different random seeds (in response to Reviewer BNoM). These results demonstrate the effectiveness and stability of the proposed pre-training framework.

---

> ### Author Response · Authors · 2025-08-02
>
> We further add the formal comparison results with positive samples from the same trial but not strictly temporally aligned, as denoted in the response to Q3.
>
> | Method         | Accuracy       | Precision      | Recall         | F1 Score       | AUROC          |
> |----------------|----------------|----------------|----------------|----------------|----------------|
> | not aligned     | 55.47 ± 1.71   | 55.17 ± 1.41   | 55.56 ± 1.70   | 54.94 ± 1.49   | 80.97 ± 1.29   |
> | aligned (Ours)  | 65.02 ± 0.98   | 65.06 ± 1.28   | 65.53 ± 0.66   | 64.85 ± 1.01   | 88.70 ± 0.98   |
>
> Without temporal alignment, the model performance drops significantly. Under the *not aligned* setting, the model achieves only 55.47% accuracy and 80.97% AUROC. This comparison validates the importance of using temporally aligned positive pairs in contrastive learning for EEG data.

---

> ### Comment · Reviewer_YfAh · 2025-08-08
> **Not bad results**
>
> Thanks for your effort, not bad results, but this is not what I interest.

---

### Official Review · Reviewer_VGNY · 2025-06-27

**Clarity:** 3
**Significance:** 2
**Originality:** 3
**Rating:** 4
**Confidence:** 4

**Summary:**

This paper presents a task-specific multi-dataset pre-training framework named mdJPT for EEG-based emotion recognition. To address challenges such as distribution shifts and label inconsistencies across datasets, the authors propose a Cross-Dataset Covariance Alignment (CDA) loss, which aligns data distributions without requiring extensive labels. In addition, they design a hybrid spatiotemporal encoder to enhance EEG signal modeling. The proposed framework improves accuracy by 6.59% over previous approaches and demonstrates strong generalization across datasets.

**Questions:**

1. The paper mentions that fine-tuning with a few target-domain subjects is needed. Has the model been evaluated in zero-shot settings? If so, what is the performance gap, and what might this suggest about potential bottlenecks in the current method?

2. Why is valence chosen as the prediction target for DEAP, rather than arousal, dominance, or a combination? Have any experiments been conducted on the other dimensions?

3. The authors state that training is conducted for 20 epochs. Could this lead to overfitting, especially with limited labeled data? No training or validation loss curves are provided—could these be added?

4. While mdJPT performs well on SEED, SEED-V, and SEED-VII, its performance drops noticeably on SEEG-IV. What factors might explain this discrepancy? How is SEEG-IV different from the other datasets?

5. Besides the CDA loss weight λ, are there other important hyperparameters in the model? How were these values selected? Was a hyperparameter search conducted?

6. The authors are encouraged to make full use of the 9-page limit to elaborate on their methodology and findings. In particular, presenting concrete case studies or examples would enhance the reader’s understanding of how the model performs across varied datasets.

7. The Related Work section would benefit from incorporating recent literature on EEG foundation models and physiological signal learning, such as Brant-2, Brant-X, and BrainBERT.

**Ethical Concerns:**

["NO or VERY MINOR ethics concerns only"]

**Final Justification:**

The main issues were solved. I think after revision, this work can reach the bottom line of acceptance, but I suggest that the AC consider this article comprehensively and combine the opinions of other reviewers to consider the decision on this article. Thank you!

**Limitations:**

Yes

**Quality:**

2

**Strengths And Weaknesses:**

### Strengths

- The paper is technically sound and clearly presents the mdJPT framework, along with key contributions including the CDA loss, a spatiotemporal encoder, and the multi-dataset pre-training strategy.

- The authors conduct thorough evaluations across multiple public EEG datasets, using cross-validation and comparison with competitive baselines. Ablation studies are also included to analyze the contributions of the ISA Loss, CDA Loss, and MLLA encoder.

- Code is made publicly available through an anonymous repository, which supports the reproducibility of the work.


### Weaknesses

- The mdJPT framework includes several components (e.g., MLLA encoder, spatiotemporal dynamics model, CDA loss), which may lead to increased model complexity, making model implementation and deployment more challenging.

- Although the framework is designed to improve generalization, it still requires a small number of subjects from the target dataset for fine-tuning, limiting its potential in zero-shot settings.

- The paper does not fully utilize the 9-page limit. There is room to provide more in-depth explanations or case studies, especially illustrating how the model generalizes across different datasets, which would strengthen the overall contribution.

- The Related Work section omits several relevant EEG and physiological signal foundation models. Important works such as Brant-2 [1], Brant-X [2], and BrainBERT [3] are highly relevant and should be discussed.

> [1] Brant-2: Foundation model for brain signals. arXiv:2402.10251
>
> [2] Brant-X: A Unified Physiological Signal Alignment Framework. KDD 2024
>
> [3] BrainBERT: Self-supervised representation learning for intracranial recordings. arXiv:2302.14367

---

> ### Author Rebuttal · Authors · 2025-07-31
>
> We sincerely thank you for your suggestions to make the results more convincing. In revision, we (i) add **zero‑shot** evaluations on unseen datasets, where mdJPT outperforms all foundation‑model baselines and broaden evidence with DEAP arousal, dominance and VA‑quadrant prediction. Interpretability results, loss curves and hyperparameter evaluations are further added. The related work and the discussion about the results are revised. These updates more convincingly, and more comprehensively, validate the model’s generalization across datasets and label regimes.
>
> ### Q1. The issue of model complexity due to multiple components.
>
> Although our model integrates multiple components, we have paid careful attention to efficiency during design. The total number of parameters remains relatively small (1.0M) compared to existing EEG foundation models (e.g., LaBram: 5.8M, EEGPT: 4.7M), as shown in Table S2. In addition, our architecture achieves faster inference speed, owing to the parameter-efficient designs of the MLLA encoder and spatiotemporal dynamics module. We believe this strikes a favorable balance between architectural expressiveness and practical usability.
>
> ### Q2. The issue of fine-tuning on the target dataset and the evaluation in zero-shot settings.
>
> To evaluate our method on real zero-shot generalization tasks, we further conducted an experiment to directly apply the pre-trained model to a new dataset without fine-tuning. We compute cosine similarity across samples and define zero‑shot accuracy by whether the top‑similar pair shares the same label. Samples from the same trial are excluded to avoid leakage. The results with SEED-V as the target dataset are shown below.
>
> **Table: Comparison of different models under zero-shot settings.**
>
> |Top1 accuracy|  SEED-V| FACED | DEAP |
> | ----------- | ------|---|---|
> | DE baseline | 43.3  |8.9|55.4|
> | mdJPT       | 52.9  |17.4|73.3|
>
> mdJPT outperforms DE baseline in the zero‑shot setting. It improves by 9.6% over the DE baseline on the SEED-V dataset. On the FACED and DEAP datasets, mdJPT also outperforms the DE baseline by 8.5% and 17.9%, respectively. We will add the results of other comparison models in the discussion session.
>
> To our knowledge, this is the first zero‑shot cross‑dataset EEG emotion experiment. Prior work typically fine‑tunes per subject. These results highlight the robustness and generalizability of our framework.
>
> ### Q3. Present concrete case studies or examples of how the model performs across varied datasets
>
> We identify the most critical features for emotion classification on multiple datasets and visualize its related temporal patterns and spatial transition patterns through the integrated gradient method. The feature importances for an emotion category are correlated between different datasets (as shown in Fig. S3), indicating a shared feature space is learned across datasets. We will further include the visualization of temporal and spatial patterns of critical features in the revised version.
>
> ### Q4. Introduce important EEG foundation models in Related Work.
>
> To present a more comprehensive review of the current foundation models, we add the introduction of BrainBERT, Brant, Brant-X, Brant-2, and CBraMod in the Related Work.
>
> > "BrainBERT implemented a masked auto-encoder on SEEG time frequency representations and improved the performance of speech decoding."
>
> > "Brant-X aligns EEG with EXG (EOG/ECG/EMG) signals through contrastive learning. It pulls the representations of simultaneous EEG and EXG patches together and pushes apart those of different time segments. Brant-2 combines EEG and SEEG data from more than 15k subjects, uses temporal and frequency features for mask-prediction and forecasting in pre-training, and demonstrates its strength in downstream tasks of seizure detection/prediction, sleep stage classification, motor movement/imagery, and emotion recognition. CBraMod designed a criss-cross attention mechanism to process temporal and spatial information separately. It used more than 27,000 hours of EEG data for pre-training and was evaluated on 10 downstream tasks. These methods demonstrate the current trend of merging larger datasets for pre-training and combining the information from multiple modalities."
>
> ### Q5. The prediction of dimensions other than valence on the DEAP dataset.
>
> In the previous submission, we prioritized a compact cross‑dataset analysis and therefore focused on valence, the dimension most commonly evaluated in previous EEG emotion studies. To evaluate the model more comprehensively,  we now add the binary prediction of high/low arousal and high/low dominance, and a four-category classification of the valence-arousal quadrants. Our model outperforms the baseline model by a large margin, demonstrating the generalization ability of the pre-trained model to new emotion rating paradigms (emotion dimensions rather than discrete categories) and its adaptability across different affective dimensions.
>
> | Evaluation          | Method      | Accuracy | Precision | Recall | F1 Score | AUROC |
> | ------------------- | ----------- | -------- | --------- | ------ | -------- | ----- |
> | **Arousal**         | DE baseline |58.67±1.27|59.00±1.60|58.67±1.27|58.36±1.21|60.67±1.97|
> |                     | mdJPT       |69.36±1.54|69.60±1.68|69.36±1.54|69.27±1.52|74.19±0.92|
> | **Dominance**       | DE baseline |55.01 ± 1.17|55.29±1.47|55.01±1.17|54.56±0.92|56.08±1.89|
> |                     | mdJPT       |73.52 ± 1.61|73.64±1.47|73.52±1.61|73.48±1.67|79.47±1.50|
> | **Valence-arousal** | DE baseline |34.03±1.06|33.81±2.84|32.59±1.21|30.04 ± 3.18|57.18 ± 1.02|
> |                     | mdJPT       |52.94±1.42|50.47±1.23|51.10±1.83|49.68±2.90|75.78 ± 1.34|
>
> ### Q6. Whether training for 20 epochs leads to overfitting? Can the loss curves be added?
>
> Specifically, we pre-train the model for 20 epochs. As the data size in pre-training is relatively large, this does not lead to overfitting. The training and validation loss generally converges after 20 epochs. In fine-tuning, the classifier is trained with early stopping. The loss curves generally converge in a few epochs and exhibit small fluctuations since then. We will also add the loss curves for pre-training and fine-tuning in the revised version.
>
> ### Q7. How to explain the performance drop on the SEED-IV dataset?
>
> To better understand the relatively lower performance on the SEED-IV dataset, we conducted a t-SNE visualization of the learned feature representations and observed that SEED-IV samples are more distant from those of other datasets in the latent space, suggesting a larger domain gap. During preprocessing, we noticed that SEED-IV recordings tend to contain more signal fluctuations and potential artifacts, such as higher variance in channel recordings and frequent noise spikes.
>
> These differences may partially account for the performance drop on this dataset. Compared to methods such as MMM, which first extract and smooth features before model training, mdJPT directly learns from EEG time series. This design prioritizes the preservation of fine-grained temporal information but may also make the model more exposed to noise in certain cases. Nevertheless, on datasets other than SEED-IV, mdJPT generally achieves stronger results than MMM, highlighting the benefits of learning directly from EEG time series. Furthermore, mdJPT consistently outperforms other foundation models that also operate on EEG time series (e.g., LaBram, EEGPT) on SEED-IV, demonstrating a degree of robustness even under challenging recording conditions.
>
> ### Q8. Evaluation of other model hyperparameters.
>
> Hyperparameters were primarily determined through established heuristics, with targeted searches conducted for several pivotal parameters. Beyond CDA weight, we found several important hyperparameters in preliminary experiments, including MLLA patch stride (which cannot be too large to enable a better extraction of local temporal features) and MLLA output dimension (needs to be at least 32 to achieve a satisfactory performance). We will report the effects of varying these hyperparameters on model performance.

---

> > ### Comment · Reviewer_VGNY · 2025-08-01
> > **Thank you**
> >
> > Thanks for your response. The main issues were solved, and I have raised my score to 4. I hope that the revisions made by the authors during the rebuttal stage will be fully incorporated into the final version of the paper.

---

> > > ### Author Response · Authors · 2025-08-07
> > >
> > > We sincerely thank you for your positive feedback. In the past few days, we further evaluated the performance of other comparison EEG foundation models on zero-shot generalization. Notably, our model consistently outperforms all comparison methods.
> > >
> > > **Table: Comparison of different models under zero-shot settings.**
> > >
> > > |Top1 accuracy|  SEED-V| FACED | DEAP |
> > > | ----------- | -------|-------|------|
> > > | DE baseline | 43.3  |8.9 |55.4|
> > > | LaBram      | 39.7  |10.2|67.0|
> > > | EEGPT       | 37.0  |11.6|62.8|
> > > | mdJPT       |**52.9**|**17.4**|**73.3**|
> > >
> > > On a challenging dataset like FACED, other models all dropped around the chance level. Only mdJPT achieves a better-than-chance accuracy. On SEED-V, mdJPT (52.9%) outperforms other models by 9.6% and even outperforms most fine-tuned models (DE baseline: 45.58%, LaBram: 41.80%, and EEGPT: 45.27%). On DEAP, mdJPT achieves a high accuracy of 73.3%, even higher than the best fine-tuned model. This could be due to that fine-tuning on only a few subjects may lead to overfitting when individual differences are large. Here, MMM is not included for zero-shot settings because it necessitates fine-tuning on a new dataset to optimize the region-wise tokens. These results demonstrate mdJPT as a promising step towards universal emotion recognition.
> > >
> > > We appreciate your constructive suggestions throughout the review process and will integrate every revision discussed in our rebuttal—including the additional experiments, clarifications, and extended analyses—into the camera-ready version.

---

### Official Review · Reviewer_CDLA · 2025-07-02

**Clarity:** 2
**Significance:** 2
**Originality:** 3
**Rating:** 4
**Confidence:** 4

**Summary:**

This paper presents a novel multi-dataset joint pre-training framework (mdJPT) for EEG-based emotion recognition, incorporating a Cross-Dataset covariance Alignment (CDA) loss and a hybrid spatiotemporal encoder. Despite demonstrating a 6.59% average F1-score improvement over existing EEG foundation models in cross-dataset generalization, several limitations need to be addressed to enhance the impact and reproducibility of this work.

1. The claim of "zero-shot generalization to new emotion categories" in Section 2.3 is not supported by the experimental results. The framework requires fine-tuning with 25% of target subjects, as described in Section 3.3, which contradicts the definition of true zero-shot learning. This limitation significantly undermines the applicability of the framework in scenarios where labeled data is scarce.

2. Residual feature distribution differences persist despite the application of CDA, as noted in Section 4. The approach struggles to handle extreme heterogeneity, such as the difference between FACED’s 9-class emotions and SEED’s 3 classes. This is evidenced by the relatively low F1-score of 20.32% on FACED, as shown in Table 3. The reliance of CDA solely on covariance may overlook higher-order distribution shifts.

3. The tuning of the CDA weight is critical but inadequately explained in Section 3.5. Performance varies non-monotonically, as observed in Table 5, where the F1-score decreases from 65.02% to 64.08% when the weight increases from 0.02 to 0.1. This lack of clear guidelines for selecting the weight parameter λ in Eq. 8 poses a risk to the reproducibility of the results.

**Questions:**

1. To evaluate the model's generalization ability, it is necessary to test zero-shot transfer learning by keeping the encoder frozen and directly assessing performance on unseen datasets. Additionally, previous claims about generalization should be reformulated to reflect the current dependency on fine-tuning, ensuring that conclusions align with reality.

2. Efforts should be made to enhance distribution alignment. This can be achieved by expanding the current CDA (Contrastive Distributional Alignment) to include higher-order statistics such as maximum mean discrepancy and incorporating domain-adversarial training strategies. To verify the effectiveness of feature distribution alignment, visualization techniques like t-SNE plots should be used to display the feature distributions before and after alignment.

**Ethical Concerns:**

["Major Concern: Data privacy, copyright, and consent"]

**Limitations:**

yes

**Paper Formatting Concerns:**

It is ok

**Quality:**

3

**Strengths And Weaknesses:**

Strengths:
1. The CDA loss effectively mitigates inter-dataset distribution shifts by aligning second-order statistical properties without requiring labels or subject-specific calibration (Section 2.3). This addresses a critical gap in EEG emotion recognition.

2.The hybrid encoder (Mamba-like linear attention + spatiotemporal dynamics) successfully captures long-term dependencies and complex EEG dynamics, outperforming vanilla transformers by >2% F1-score (Table 4).

Weaknesses:
1. The claim of "zero-shot generalization to new emotion categories" in Section 2.3 is not supported by the experimental results. The framework requires fine-tuning with 25% of target subjects, as described in Section 3.3, which contradicts the definition of true zero-shot learning. This limitation significantly undermines the applicability of the framework in scenarios where labeled data is scarce.

2. Residual feature distribution differences persist despite the application of CDA, as noted in Section 4. The approach struggles to handle extreme heterogeneity, such as the difference between FACED’s 9-class emotions and SEED’s 3 classes. This is evidenced by the relatively low F1-score of 20.32% on FACED, as shown in Table 3. The reliance of CDA solely on covariance may overlook higher-order distribution shifts.

3. The tuning of the CDA weight is critical but inadequately explained in Section 3.5. Performance varies non-monotonically, as observed in Table 5, where the F1-score decreases from 65.02% to 64.08% when the weight increases from 0.02 to 0.1. This lack of clear guidelines for selecting the weight parameter λ in Eq. 8 poses a risk to the reproducibility of the results.

---

> ### Author Rebuttal · Authors · 2025-07-31
>
> We sincerely thank you for the thoughtful review and for recognizing our performance gains. In the revision, we add true **zero‑shot** evaluations on unseen datasets, where mdJPT outperforms all foundation‑model baselines. We also assess higher‑order alignment via MK‑MMD (Gaussian, multi‑kernel) and find CDA remains competitive while being simpler to optimize. We further include t‑SNE and clustering‑coefficient analyses to demonstrate the pre-training effect. We appreciate your suggestions—they substantially improved the clarity and completeness of the work.
>
> ### Q1. The limitation of fine-tuning on the target dataset.
>
> To evaluate our method on real zero-shot generalization tasks, we further conducted an experiment to directly apply the pre-trained model to a new dataset without fine-tuning. We compute cosine similarity across samples and define zero‑shot accuracy by whether the top‑similar pair shares the same label. Samples from the same trial are excluded to avoid leakage. The results with SEED-V as the target dataset are shown below.
>
> **Table: Comparison of different models under zero-shot settings.**
>
> |Top1 accuracy|  SEED-V| FACED | DEAP |
> | ----------- | ------|---|---|
> | DE baseline | 43.3  | 8.9 | 55.4|
> | mdJPT       | 52.9  |17.4 | 73.3|
>
> mdJPT outperforms DE baseline in the zero‑shot setting. It improves by 9.6% over the DE baseline on the SEED-V dataset. On the FACED and DEAP datasets, mdJPT also outperforms the DE baseline by 8.5% and 17.9%, respectively. We will add the results of other comparison models in the discussion session.
>
> We clarify the original evaluation protocol as "few-subject fine-tuning", and the new protocol as "zero-shot generalization".
>
> To our knowledge, this is the first zero‑shot cross‑dataset EEG emotion experiment. Prior work typically fine‑tunes per subject. These results highlight the robustness and generalizability of our framework.
>
> ### Q2.1 Residual feature distribution differences persist despite the application of CDA. The reliance of CDA solely on covariance may overlook higher-order distribution shifts.
>
> We understand the reviewer's concern about the possibility of higher-order distribution shifts. We agree that relying only on covariance may miss higher‑order distribution differences. Indeed, CDA loss aligns global second-order statistics (i.e., inter-channel covariance structures) across subjects and datasets. This captures **inter-region coupling** (or functional connectivity) patterns in EEG—features that have been increasingly emphasized in recent neuroscience and EEG decoding literature [1-3].
>
> Although CDA can align EEG with the second-order relations, residual distribution shifts may persist, especially under extreme heterogeneity. To test the impact of higher-order distribution, we add an experiment with **higher-order alignment** using Multiple Kernel Maximum Mean Discrepancy loss (MK-MMD, Gaussian kernel). The MMD loss with a Gaussian kernel includes all moments of the distribution. It implicitly maps inputs into an infinite-dimensional reproducing kernel Hilbert space (RKHS), where the distance between distributions captures differences across all orders of moments. Multiple kernels (five here) are introduced to reduce the reliance on the Gaussian kernel length. All other settings are kept the same as our models. The results with SEED-V as the target dataset are shown below:
>
> **Table: Comparison with MK-MMD loss.**
>
> | Method          | Accuracy | Precision | Recall | F1 Score | AUROC |
> | --------------- | -------- | --------- | ------ | -------- | ----- |
> | MK-MMD loss     |64.33±1.24|64.52±1.86|64.77±1.22|64.00±1.40|87.73±0.89|
> | CDA loss (ours) |**65.02±0.98**|**65.06±1.28**|**65.53±0.66**|**64.85±1.01**|**88.70±0.98**|
>
> CDA slightly outperforms MK-MMD in our setting, suggesting that aligning second-order statistics already captures much of the cross-dataset shift in EEG (e.g., montage/linear mixing). Higher-order criteria like MK-MMD can, in principle, model richer differences, but in our experiments they did not yield clear gains. The residual discrepancies likely arise from shifts not easily addressed by distribution matching alone (e.g., elicitation protocol mismatch or pronounced nonstationarity), so even higher-order matching may not fully close the gap. Besides, kernel-based higher-order alignment adds tuning complexity and can be harder to optimize stably. We will explore complementary solutions in future work.
>
> ### Q2.2 The effectiveness of feature distribution alignment should be visualized.
>
> In **Appendix D.2**, we visualized the latent representations on each target dataset. Although emotion labels were not used in pre-training, the features extracted by pre-trained mdJPT are visually more separable than the original DE features, especially on datasets with fewer emotions, like SEED. This demonstrates the effectiveness of pre-training in facilitating downstream emotion recognition tasks. In the revised version, we will provide the t-sne visualization of all datasets before and after pre-training, to investigate whether pre-training can effectively mitigate the data discrepancy across datasets.
>
> ### Q3. The tuning of the CDA weight not inadequately explained in Section 3.5.
>
> We agree that performance varies with λ and is not strictly monotonic. In **Table 5**, the drop at λ=0.01 coincides with a larger STD, which could be attributed to occasional training instability when ISA and CDA losses exert partially opposing gradients at this λ. In the revision, we will run a denser sweep over different λs (λ∈{0, 0.001, 0.002, 0.005, 0.0075, 0.010, 0.015, ...}) and repeat each point over multiple random seeds, providing a clear, reproducible guideline for selecting λ.

---

> > ### Comment · Reviewer_CDLA · 2025-08-03
> >
> > Thank you for your efforts. This response has already partially addressed my concerns.

---

> > > ### Author Response · Authors · 2025-08-07
> > >
> > > Thank you very much for your feedback. We have carried out further analyses to (i) evaluate leading EEG foundation models under the identical zero-shot protocol, showing that mdJPT consistently outperforms them and (ii) provide a clearer, quantitative picture of how pre-training narrows cross-dataset gaps.
> > >
> > > ### **Comparison with other EEG foundation models on the zero-shot setting**
> > >
> > > We add the comparison with other EEG foundation models on the zero-shot setting. mdJPT consistently outperforms these models on three representative datasets. On a challenging dataset like FACED, other models all dropped around the chance level. Only mdJPT achieves a better-than-chance accuracy. On SEED-V, mdJPT outperforms other models by 9.6% and even outperforms most fine-tuned models (DE baseline: 45.58%, LaBram: 41.80%, and EEGPT: 45.27%). On DEAP, mdJPT achieves a high accuracy of 73.3%, even higher than the best fine-tuned model. This could be due to that fine-tuning on only a few subjects may lead to overfitting when individual differences are large. Here, MMM is not included for zero-shot settings because it necessitates fine-tuning on a new dataset to optimize the region-wise tokens. These results demonstrate mdJPT as a foundational step towards the zero-generalization to new datasets with new emotion categories.
> > >
> > > **Table 1: Comparison of different models under zero-shot settings.**
> > >
> > > |Top1 accuracy|  SEED-V| FACED | DEAP |
> > > | ----------- | -------|-------|------|
> > > | DE baseline | 43.3  |8.9 |55.4|
> > > | LaBram      | 39.7  |10.2|67.0|
> > > | EEGPT       | 37.0  |11.6|62.8|
> > > | mdJPT       |**52.9**|**17.4**|**73.3**|
> > >
> > > ### **Quantification of cross-dataset divergence and the effect of pre-training**
> > >
> > > To give a quantitative view of how mdJPT reduces domain gaps, we computed Silhouette scores on t-SNE embeddings before (raw DE features, Table 2) and after pre-training (Table 3). The score is calculated with the dataset label as the cluster indicator: the lower the value, the better samples from different datasets mix.
> > >
> > > Two key observations can be summarized from comparing Table 3 and Table 2: (i) Global reduction of domain gaps. After pre-training, nearly every pairwise Silhouette value drops—sometimes by an order of magnitude—showing that mdJPT draws heterogeneous datasets into a shared latent space. (ii) Tougher domains remain identifiable. The highest residual gaps occur among FACED, SEED-VII, and EmoEEG-imagery (the imagery-induced set we recently added for completeness). These cases motivate future work on even richer alignment objectives.
> > >
> > > Overall, this analysis clarifies why mdJPT transfers well across most datasets yet still leaves room for improvement on the most fine-grained or protocol-divergent scenarios, aligning with the performance patterns reported in the main text.
> > >
> > > **Table 2: Silhouette scores of DE features between datasets.**
> > >
> > > |Silhouette score |  EmoEEG-imagery |FACED| DEAP| SEED |SEED-V | SEED-VII|
> > > | --- | ------         |---  | --- |---   |---    |---      |
> > > | **EmoEEG-imagery** |  \ |0.216|0.059|0.219 |0.166  | 0.107   |
> > > | **FACED** |       \       |  \  |0.248|-0.039|0.063  | 0.007   |
> > > | **DEAP**  |        \      |   \ |  \   |0.198 |0.147  | 0.100   |
> > > | **SEED**  |        \      |   \  |   \  |   \   |0.050  | 0.098   |
> > > | **SEED-V**|       \       |   \  |  \  | \    |   \    | 0.068   |
> > >
> > > **Table 3: Silhouette scores of latent representations between datasets.**
> > >
> > > |Silhouette score |  EmoEEG-imagery |FACED| DEAP| SEED |SEED-V | SEED-VII|
> > > | --- | ------         |---  | --- |---   |---    |---      |
> > > | **EmoEEG-imagery** |  \   |0.057|0.067|0.087 |0.062  | 0.126   |
> > > | **FACED** |     \         |  \   |0.056|0.020 |0.043  | 0.167   |
> > > | **DEAP**  |      \        |   \  |   \  |0.007 |-0.001 | 0.056   |
> > > | **SEED**  |      \        |  \   |  \   |  \    |0.009  | 0.070   |
> > > | **SEED-V**|      \        |   \  |   \  |  \    |  \     | 0.048   |
> > >
> > > We are happy to address any additional questions or suggestions you may have. Thank you again for your invaluable comments!

---

### Official Review · Reviewer_BNoM · 2025-07-03

**Clarity:** 3
**Significance:** 4
**Originality:** 3
**Rating:** 4
**Confidence:** 4

**Summary:**

The authors propose mdJPT (Multi‑dataset Joint Pre‑Training), a task‑specific framework for EEG‑based emotion recognition. The core Idea is to jointly pre‑train a hybrid EEG encoder on several emotion datasets, while aligning second‑order statistics (Cross‑Dataset Alignment, CDA) and inter‑subject representations (Inter‑Subject Alignment, ISA). The architecture employs a hybrid encoder combining a lightweight Mamba-style linear attention module and a spatiotemporal dynamics block. Evaluation across six public EEG datasets using a leave-one-dataset-out protocol shows that mdJPT significantly outperforms both traditional models and recent EEG foundation models, improving average F1-scores by 6.6 percentage points.

**Questions:**

1) It is unclear how much labeled data from the held-out (target) dataset is used for fine-tuning. The paper mentions "fine-tuning" but does not specify: How many subjects or trials per subject are used?
2) Whether experiments were repeated over multiple random seeds. Whether splits were consistent across models.
3) The paper does not specify whether the same EEG preprocessing was used for the baseline.
4) Why does the baseline not include any task-specific cross-dataset model? The authors should add them or discuss why these baselines were chosen.
5) Why is the EEG dataset from other elicitation methods not used for experiments? Please discuss.
6) The authors should discuss the dataset results, and why the results vary so much.

**Ethical Concerns:**

["NO or VERY MINOR ethics concerns only"]

**Final Justification:**

The authors have addressed most points well in the rebuttal, and I appreciate their detailed response. However, I will maintain my original score, as the work would benefit from a deeper discussion on the intuition of the mdJPT method and results at the data level, particularly to support generalizability claims.

**Limitations:**

Yes, the paper has a limitation section, but the paper does not explicitly address the potential negative societal impacts of EEG-based emotion recognition. Further, the paper would benefit from addressing the lack of diversity in the datasets (age, cultural background, mental health status), which limits generalisability to real-world, heterogeneous populations. The authors should also consider discussing risks such as misuse in surveillance or workplace monitoring, overinterpretation of emotional states, and the ethical concerns around applying such models without informed consent. Also, given EEG hardware is bulky, their real-world applicability would be limited; this should also be added.

**Paper Formatting Concerns:**

Some figures lack detailed captions explaining all subcomponents or labels, requiring the reader to search the main text for clarification.

**Quality:**

3

**Strengths And Weaknesses:**

Strength:
1) Solves the problem of data scarcity by focusing on the pre-tarining with multiple datasets. Moreover, it combines task‑specific multi‑dataset pre‑training with covariance‑centric alignment; prior EEG foundation models focus on task‑agnostic breadth.
2) The paper addresses a key challenge in EEG-based emotion recognition: poor cross-dataset generalisation. By pretraining across multiple datasets with targeted alignment strategies, mdJPT offers a meaningful step forward for real-world deployment also since combining dataset adds to data diversity in training.
3) The integration of second-order statistical alignment (CDA) and subject-level contrastive learning (ISA) is interesting.
4) The paper is clearly written, with well-structured sections.

Weakness:
1) The assumption that EEG emotion datasets are misaligned is taken from prior knowledge, but the paper does not directly test or visualise this. I think including visual evidence would make the motivation for their alignment techniques much more compelling.
2) All datasets are video-evoked emotion EEG data collected in lab-settings, no dataset from other elicitation technique is used like, semi-naturalistic tasks (public speaking, debate, memory recall or others)
3) Author should discuss the lower accuracy and poor result on dataset like FACED - 23%.
4) While subject-independent testing is performed, the method still assumes access to labeled data from some subjects in the target dataset for fine-tuning. No zero-shot results are presented.
5) Exact subject split ratios are missing (e.g., how many subjects used for fine-tuning vs. test?). The number of trials per subject used for training vs. testing is unclear.
6) The number of random seeds or runs per dataset is not stated clearly, though metrics are reported as mean ± SD.

---

> ### Author Rebuttal · Authors · 2025-07-31
>
> # Global response
>
> We sincerely thank the reviewers for the thoughtful and constructive feedback. We are encouraged by the recognition of our framework's significance and competitive performance. At the same time, we acknowledge the concerns regarding our motivation, methodology, and clarity, and we are committed to addressing all points thoroughly in our revision. We also provide additional experiments and explanations to address reviewers' concerns:
>
> 1. The performance of zero-shot generalization to unseen datasets. (Reviewer **BNoM**, **CDLA**, and **VGNY**)
>
> 2. The performance on emotion-elicitation protocols other than video-elicitation. (Reviewer **BNoM**)
>
> 3. Evaluation on more affective dimensions on the DEAP dataset and the transfer from DEAP to other datasets. (Reviewer **VGNY** and **YfAh**)
>
> 4. Comparison with alternative methods, including supervised contrastive learning and MMD loss. (Reviewer **CDLA** and **YfAh**)
>
> 5. The motivation behind the design of CDA loss and MLLA encoder. (Reviewer **YfAh**)
>
> Below, we respond to each reviewer in detail.
>
> ## Response to reviewer BNoM
>
> We sincerely thank you for the thoughtful and constructive review. We are glad you found the problem setting and our multi‑dataset pre-training framework meaningful. Your comments directly helped us improve the paper’s clarity, rigor, and scope. Following the suggestions, we added a **zero‑shot** evaluation with no fine‑tuning on the target dataset and included results on the `EmoEEG‑MC imagery‑induced dataset` to test generalization **beyond video stimuli**. We add analyses on dataset misalignment and the effect of random seeds, clarify method details, and add the discussion of results and limitations.
>
> ### Q1. Visualization of dataset misalignment.
>
> We visualize t‑SNE embeddings of differential entropy (DE) features extracted from EEG signals across datasets. Data from the same dataset cluster together, while different datasets are separated. This demonstrates large cross‑dataset misalignment in EEG features. We will add the figure in the revised version.
>
> ---
>
> ### Q2. No dataset from elicitation techniques other than video-evoking is used, like semi-naturalistic tasks.
>
> Most widely used EEG emotion datasets are video‑induced, which emphasizes controlled, video‑elicited designs. To assess generalizability beyond video stimuli, we add experiments on `EmoEEG‑MC` [1], a multi‑context dataset published in 2025 that includes an imagery‑induced paradigm (guided narratives with active imagination), eliciting more internally driven and sustained emotions.
>
> We evaluate on the imagery task using the same protocol: pretrain on six video‑induced datasets, then fine‑tune on 1/4 of EmoEEG‑MC subjects and test on the remaining.
>
> **Table: Comparison of different models on the EmoEEG-MC dataset with imagery-induced emotion.**
>
> | Method      | Accuracy | Precision | Recall | F1 Score | AUROC |
> | ----------- | -------- | --------- | ------ | -------- | ----- |
> | DE baseline |18.38±1.12|17.91±1.62|18.38±1.12|17.87±1.37|54.11±1.36|
> | mdJPT       |20.56±0.79|20.83±0.66|20.56±0.79|19.85±2.11|56.14±0.78|
>
> Our method outperforms other baselines in the imagery-induced setting. Comparison with other models will be provided in the discussion session. Transfer from video to imagery remains relatively low, likely due to scenario discrepancies. Future work needs to pretrain on more diverse and naturalistic tasks.
>
> \[1] Xu X, Shen X, Chen X, et al. *A Multi‑Context Emotional EEG Dataset for Cross‑Context Emotion Decoding*. Scientific Data, 2025, 12(1): 1142.
>
> ---
>
> ### Q3. Authors should discuss the lower accuracy on datasets like FACED. Why do the results vary so much?
>
> Performance varies for several reasons. First, label space differs: DEAP (2 categories of high/low valence), SEED (3 categories), SEED‑IV (4 categories), SEED‑V (5 categories), SEED‑VII (7 categories), and FACED (9 fine‑grained emotions, with more positive categories). Several positive emotions in FACED (joy, amusement, inspiration, tenderness) may be subtle in neural differences and harder to separate. Second, annotation standards differ: DEAP uses continuous affective ratings, while others use discrete labels, affecting uncertainty and separability.
>
> Sensor setup and signal quality also matter. SEED uses 62 channels; DEAP and FACED use 32, reducing spatial resolution. During preprocessing, SEED‑IV showed more noise/artifacts (higher channel variance, more disruptions). A t‑SNE visualization places SEED‑IV farther from other datasets, indicating a bigger domain gap, which may explain its relatively lower performance.
>
> ---
>
> ### Q4. The issue of fine-tuning and the evaluation of zero-shot performance.
>
> To evaluate our method on real zero-shot generalization tasks, we further conducted an experiment to directly apply the pre-trained model to a new dataset without fine-tuning. We compute cosine similarity across samples and define zero‑shot accuracy by whether the top‑similar pair shares the same label. Samples from the same trial are excluded to avoid leakage. The results with SEED-V as the target dataset are shown below.
>
> **Table: Comparison of different models under zero-shot settings.**
>
> |Top1 accuracy|  SEED-V| FACED | DEAP |
> | ----------- | ------|---|---|
> | DE baseline | 43.3  |8.9|55.4|
> | mdJPT       | 52.9  |17.4|73.3|
>
> mdJPT outperforms DE baseline in the zero‑shot setting. It improves by 9.6% over the DE baseline on the SEED-V dataset. On the FACED and DEAP datasets, mdJPT also outperforms the DE baseline by 8.5% and 17.9%, respectively. We will add the results of other comparison models in the discussion session.
>
> To our knowledge, this is the first zero‑shot cross‑dataset EEG emotion experiment. Prior work typically fine‑tunes per subject. These results highlight the robustness and generalizability of our framework.
>
> ---
>
> ### Q5. Exact subject split ratios for fine-tuning vs. test.
>
> In lines 185–187: “We split the target dataset at a 1:3 subject ratio for fine‑tuning and testing. A leave‑one‑dataset‑out cross‑validation is employed. Our method is pre-trained on all datasets except the target.” Concretely, 25% of target subjects are randomly selected for fine‑tuning and 75% for testing. We repeat the random split six times and report mean ± SD. We will clarify this in Section 3.1.
>
> ---
>
> ### Q6.1 The number of random seeds or runs per dataset.
>
> For fine‑tuning/testing, we use six random splits (as in Q5) and report mean ± SD. During pre-training, the seed (19260832) was randomly chosen without cherry‑picking. We will replicate the full pipeline multiple times with different random seeds and report their consistency in the discussion session.
>
> ---
>
> ### Q6.2 Whether splits were consistent across models?
>
> Yes. We strictly use the same fine‑tuning/test splits and the same seeds for all models.
>
> ---
>
> ### Q7. Whether the same EEG preprocessing was used for the baseline?
>
> Yes. All methods share the same standard EEG preprocessing. Details are in Appendix C.
>
> ---
>
> ### Q8. Why does the baseline not include any task-specific cross-dataset model?
>
> Our baselines match mdJPT’s goals: (i) multi‑dataset joint pretraining, (ii) generalization to unseen categories, and (iii) subject‑independent evaluation (apply to new subjects directly). Most cross‑dataset methods are one‑to‑one adaptations, assuming the same label space and requiring target data—often per‑subject—for training/fine‑tuning (e.g., adversarial alignment). As summarized in Table 1, these pipelines do not simultaneously handle new categories and new subjects, the core challenge addressed by our framework.
>
> For fair comparison, we use foundation‑model baselines, pretrained on multiple large datasets and evaluated on new ones, under the same lightweight classifier protocol. We will clarify this rationale in the revision.
>
> ---
>
> ### Q9. Limitations of generalizability, negative societal effects, and the real-world applicability.
>
> We will add the following paragraphs to the Discussion:
>
> >“Another limitation is the lack of diversity in current EEG emotion datasets—participants are often from limited age groups (generally young people), cultural backgrounds (Probably British for DEAP and Chinese for other datasets), and mental health status (healthy), limiting generalizability to heterogeneous populations. Real‑world deployment also remains constrained by the practicality of current EEG hardware, which is often cumbersome. The ongoing development of lightweight, wearable EEG is expected to enable more practical applications.
>
> >We also acknowledge societal and ethical risks, including potential misuse in surveillance, workplace monitoring, and unauthorized emotional profiling. Such uses may lead to overinterpretation or coercion without informed consent. As the field advances, it is essential to establish clear regulatory oversight and develop privacy‑preserving frameworks.”
>
>
>
> ### Q10. Some figures lack detailed captions explaining all subcomponents or labels.
>
> We will revise the figure captions to include more details. For example, in Figure 2, we revise the caption of sub-figure A as: "A) The overall framework. Our framework includes three stages: multi-dataset joint pre-training, classifier fine-tuning, and testing. The EEG encoder contains an MLLA channel encoder and a spatiotemporal dynamics model. During pre-training, the encoder is optimized with a combination of CDA and inter-subject alignment (ISA) losses. During fine-tuning, the encoder is frozen, and a lightweight classifier is trained on a small subset of subjects from the target dataset. Evaluation is then performed on the remaining (held-out) subjects.

---

> ### Comment · Reviewer_BNoM · 2025-08-01
> **Response to Rebuttal**
>
> Thank you for the rebuttal. I appreciate the authors’ efforts to incorporate additional experiments and provide clarifications, especially the zero-shot evaluation, t-SNE visualization, and inclusion of the EmoEEG-MC imagery-induced dataset. However, I still believe that some results, particularly the low performance on datasets (FACED and EmoEEG-MC) and zero-shot transferability, still need a deeper discussion at the data level, given that the core claim of the work is to improve generalization across datasets. Overall, this is a promising and timely contribution to EEG pertaining.

---

> > ### Author Response · Authors · 2025-08-07
> >
> > We sincerely thank you for your constructive feedback. Motivated by your suggestion for a deeper data-level discussion, we undertook three additional investigations: (i) we quantified cross-dataset divergence via Silhouette scores, demonstrating the effect of pre-training; (ii) we reported full confusion matrices for both datasets, revealing the precise fine-grained confusions that drive the lower scores; and (iii) we evaluated leading EEG foundation models under the identical zero-shot protocol, showing that mdJPT consistently outperforms them. These results demonstrate mdJPT as a promising step towards universal emotion recognition, while highlighting the need for larger, more diverse datasets and further algorithmic refinements to tackle more challenging scenarios.
> >
> > ### **Quantification of cross-dataset divergence and the effect of pre-training**
> >
> > To better understand the discrepancy across datasets, we computed Silhouette scores on t-SNE embeddings before (raw DE features, Table 1) and after pre-training (Table 2). Silhouette score is calculated with the dataset label as the cluster indicator: the lower the value, the better samples from different datasets mix. We can see that for differential entropy (DE) features (Table 1), EmoEEG-imagery has relatively large Silhouette scores with other datasets, indicating samples from EmoEEG-imagery are more separated from other datasets. The discrepancy is generally small within SEED-series datasets. These results demonstrate that the EEG data of imagery-induced emotion have a large difference from video-induced emotion, which could explain the difficulty of transfer to an imagery scenario.
> >
> > By comparing Table 2 to Table 1, we can see that after pre-training, the Silhouette score of latent representations generally drops—sometimes by an order of magnitude—showing that mdJPT draws heterogeneous datasets into a shared latent space. The difference between the EmoEEG-imagery dataset and video-induced datasets is mitigated. However, there remains a discrepancy between certain datasets, especially EmoEEG-imagery, FACED, and SEED-VII. These datasets with more fine-grained emotion categories and imagery-induction context may be more difficult for the model to align. This analysis clarifies why mdJPT transfers well across most datasets, yet still leaves room for improvement on the most fine-grained or protocol-divergent scenarios.
> >
> > **Table 1: Silhouette scores of DE features between datasets.**
> >
> > |Silhouette score |  EmoEEG-imagery |FACED| DEAP| SEED |SEED-V | SEED-VII|
> > | --- | ------         |---  | --- |---   |---    |---      |
> > | **EmoEEG-imagery** |  \ |0.216|0.059|0.219 |0.166  | 0.107   |
> > | **FACED** |       \       |  \  |0.248|-0.039|0.063  | 0.007   |
> > | **DEAP**  |        \      |   \ |  \   |0.198 |0.147  | 0.100   |
> > | **SEED**  |        \      |   \  |   \  |   \   |0.050  | 0.098   |
> > | **SEED-V**|       \       |   \  |  \  | \    |   \    | 0.068   |
> >
> > **Table 2: Silhouette scores of latent representations between datasets.**
> >
> > |Silhouette score |  EmoEEG-imagery |FACED| DEAP| SEED |SEED-V | SEED-VII|
> > | --- | ------         |---  | --- |---   |---    |---      |
> > | **EmoEEG-imagery** |  \   |0.057|0.067|0.087 |0.062  | 0.126   |
> > | **FACED** |     \         |  \   |0.056|0.020 |0.043  | 0.167   |
> > | **DEAP**  |      \        |   \  |   \  |0.007 |-0.001 | 0.056   |
> > | **SEED**  |      \        |  \   |  \   |  \    |0.009  | 0.070   |
> > | **SEED-V**|      \        |   \  |   \  |  \    |  \     | 0.048   |

---

> > > ### Comment · Reviewer_BNoM · 2025-08-09
> > >
> > > Thank you for the response. Please add these changes to the final version, as these will further improve the discussion of results.

---

> > > > ### Author Response · Authors · 2025-08-09
> > > >
> > > > Thank you for your valuable comments! We will incorporate all the analyses into the final version.

---

> > ### Author Response · Authors · 2025-08-07
> >
> > ### We further add **the confusion matrices of FACED and EmoEEG-imagery:**
> >
> > To investigate the detailed confusion patterns on FACED and EmoEEG-imagery, we calculated the confusion matrices on these datasets. Each row represents a true label, and each column represents a predicted label. On the FACED dataset, the confusion matrix shows a diagonal dominance: within each row, the diagonal entry is the highest, meaning the model is more likely to predict the true label than any other single label. Among all categories, disgust, tenderness, and neutral have the highest classification accuracy, indicating they have more discriminative neural features. Joy tends to be confused with amusement or inspiration, indicating more similar neural representations across these fine-grained positive emotion categories.
> >
> > **Table 3: The confusion matrix of emotion classification on FACED.**
> >
> > |Emotions | Joy | Amusement | Inspiration | Tenderness | Neutral | Sadness | Fear | Disgust | Anger |
> > | ----------- | ------|---|---|---|---|---|---|---|---|
> > | **Joy**         |  **0.17** |  0.17 |  0.15 |  0.06 |  0.06 |  0.14 |  0.10 |  0.08 |  0.13 |
> > | **Amusement**   |  0.13 |  **0.23** |  0.09 |  0.09 |  0.07 |  0.05 |  0.11 |  0.07 |  0.09 |
> > | **Inspiration** |  0.16 |  0.07 |  **0.18** |  0.08 |  0.12 |  0.08 |  0.10 |  0.10 |  0.12 |
> > | **Tenderness**  |  0.09 |  0.08 |  0.07 |  **0.31** |  0.08 |  0.09 |  0.09 |  0.05 |  0.11 |
> > | **Neutral**    |  0.08 |  0.09 |  0.11 |  0.11 |  **0.29** |  0.09 |  0.07 |  0.10 |  0.08 |
> > | **Sadness**     |  0.11 |  0.10 |  0.05 |  0.09 |  0.13 |  **0.18** |  0.10 |  0.08 |  0.10 |
> > | **Fear**       |  0.10 |  0.10 |  0.11 |  0.09 |  0.07 |  0.11 |  **0.20** |  0.12 |  0.11 |
> > | **Disgust**    |  0.05 |  0.07 |  0.11 |  0.08 |  0.08 |  0.13 |  0.12 |  **0.33** |  0.03 |
> > | **Anger**      |  0.10 |  0.09 |  0.13 |  0.10 |  0.08 |  0.13 |  0.11 |  0.07 |  **0.22** |
> >
> > On the EmoEEG-imagery dataset, neutral achieves the highest classification accuracy and is rarely confused with other emotions. The next best-classified emotions are tenderness and fear. This result indicates that on very challenging transfer to imagery context, the model can capture some prominent distinctiveness between emotion states, such as neutral and other evoked emotions. For more fine-grained emotion categories, the direct transfer accuracy is relatively low, which calls for pre-training the model with more diverse datasets. On both FACED and EmoEEG-imagery datasets, tenderness has a relatively high accuracy, indicating a stable neural representation of tenderness.
> >
> > **Table 4: The confusion matrix of emotion classification on EmoEEG-imagery.**
> >
> > |Emotions | Joy | Inspiration | Tenderness | Neutral | Sadness | Fear | Disgust |
> > | ----------- | ------|---|---|---|---|---|---|
> > | **Joy**         |  0.15 |  0.18 |  0.15 |  0.09 |  0.12 |  0.15 |  0.17 |
> > | **Inspiration** |  0.15 |  0.16 |  0.15 |  0.10 |  0.17 |  0.15 |  0.12 |
> > | **Tenderness**  |  0.16 |  0.16 |  **0.21** |  0.08 |  0.15 |  0.13 |  0.11 |
> > | **Neutral**     |  0.10 |  0.08 |  0.10 |  **0.38** |  0.11 |  0.11 |  0.13 |
> > | **Sadness**     |  0.14 |  0.17 |  0.14 |  0.11 |  0.16 |  0.13 |  0.15 |
> > | **Fear**        |  0.13 |  0.14 |  0.13 |  0.14 |  0.15 |  **0.17** |  0.15 |
> > | **Disgust**     |  0.18 |  0.11 |  0.13 |  0.10 |  0.15 |  0.17 |  0.17 |

---

### Note · Authors · 2025-08-12

We sincerely thank all reviewers for their constructive and insightful feedback, which is invaluable in improving our work.

This paper introduces mdJPT, a task-specific multi-dataset joint pre-training paradigm for EEG-based emotion recognition, addressing inter-dataset distribution shifts, inconsistent emotion categories, and inter-subject variability. It combines a principled alignment strategy—global second-order covariance alignment and fine-grained inter-subject alignment—with a physiologically plausible, efficient EEG encoder to enable robust cross-dataset generalization.

In direct response to reviewer suggestions, we:

**Expanded evaluation** to include zero-shot cross-dataset emotion recognition on additional datasets (e.g., EmoEEG-MC imagery-induced data) and extended DEAP to multi-dimensional affective labels, confirming robust cross-dataset and cross-paradigm transfer.

**Enhanced analysis** of generalization challenges via Silhouette score divergence quantification and fine-grained confusion matrices, validating the effectiveness of pre-training in dataset alignment and identifying persistent difficulties in highly heterogeneous settings.

**Clarified methodology** by detailing the CDA loss rationale, MLLA encoder design, and necessity of temporally aligned positives in ISA loss; compared the proposed method with higher-order alignment and supervised contrastive learning.

**Strengthened ethical discussion** with explicit consideration of consent scope, privacy safeguards, demographic coverage, and potential misuse.

We believe these additions strengthen both the technical rigor and the practical transparency of the work, and we are grateful for the reviewers’ role in shaping a more robust and impactful contribution.

---

### Decision · Program_Chairs · 2025-09-17

**Decision:**

Accept (poster)

**Comment:**

This paper addresses a task-specific multi-datatset joint pre-training framework for cross-dataset emotion recognition. The key idea is to jointly pre-train a hybrid EEG encoder on several emotion datasets, introducing a cross-dataset covariance alignment and inter-subject alignment. All of reviewers feel that the approach taken in this paper is sound and promising for tackling task-specific complex problems  of EEG recognition. One of critical concerns raised by reviewers is about “zero-shot generalization to new emotion categories”. Experiments on zero-shot generalization was done during the rebuttal period and reasonable empirical results resolved the concern. The authors did a good job in responding to the issues pointed by reviewers during the rebuttal period. This made us to reach consensus on this paper.